# AI-GENERATED FACES INFLUENCE GENDER STEREO-TYPES AND RACIAL HOMOGENIZATION

## ABSTRACT

Text-to-image generative AI models such as Stable Diffusion are used daily by millions worldwide. However, the extent to which these models exhibit racial and gender stereotypes is not yet fully understood. Here, we document significant biases in Stable Diffusion across six races, two genders, 32 professions, and eight attributes. Additionally, we examine the degree to which Stable Diffusion depicts individuals of the same race as being similar to one another. This analysis reveals significant racial homogenization, e.g., depicting nearly all middle eastern men as dark-skinned, bearded, and wearing a traditional headdress. We then propose debiasing solutions that address the above stereotypes. Finally, using a preregistered experiment, we show that being presented with inclusive AI-generated faces reduces people's racial and gender biases, while being presented with non-inclusive ones increases such biases. This persists regardless of whether the images are labeled as AI-generated. Taken together, our findings emphasize the need to address biases and stereotypes in AI-generated content.

## 1    INTRODUCTION

Artificial intelligence biases refer to the systematic and unfair preferences or prejudices embedded in AI systems, often reflecting the biases present in the data used to train these systems Christian (2020). Such biases can perpetuate and even exacerbate societal inequalities, as the AI algorithms may inadvertently discriminate against certain groups. One glaring example of AI bias is the COM-PAS algorithm, which has been shown to produce racially biased predictions when assessing the likelihood of recidivism Angwin et al. (2016); Dressel & Farid (2018). Another notable example of AI bias is found in face recognition algorithms, which have been shown to discriminate based on race and gender Buolamwini & Gebru (2018). Another such example is Amazon's endeavor to use an algorithm to evaluate job candidates based on their resumes, which inadvertently penalized resumes that included phrases associated with women Dastin (2018). In the context of NLP, notable biases have been reported in popular word embedding models such as BERT and GPT-2, which associate certain occupations or stereotypes more strongly with one gender or racial group than another Nadeem et al. (2021).

In this paper, we focus on racial and gender stereotypes in SDXL (Stable Diffusion XL) Podell et al. (2023), one of the most popular text-to-image generative models used daily by millions worldwide Fatunde & Tse (2022). Recent studies have demonstrated that Stable Diffusion underrepresents certain races or genders Bianchi et al. (2023); Wang et al. (2023); Ghosh & Caliskan (2023), but none of these studies proposed debiasing solutions. Moreover, none of them offered a comprehensive examination of such biases across racial groups, genders, professions, and attributes. Other studies have proposed debiasing solutions Zhang et al. (2023a); Friedrich et al. (2023), but these are either not automated, or are unable to generate images that adequately represent complex prompts; see the Related Work section for more details. Another form of stereotype that has been overlooked in the literature is when individuals of the same race are depicted as being too similar to one another, e.g., when Middle Eastern men are all depicted as being bearded and dark-skinned, or when Middle Eastern women are all depicted as wearing a traditional headcover, or "hijab". Consequently, it remains unclear whether such homogenization (if it exists) can be addressed by diversifying the facial features of same-race individuals. Other open questions that have not been addressed to date are whether being exposed to AI-generated faces can affect people's racial and gender biases.

To address these questions, we start off by developing a classifier to predict race and gender, allowing us to quantify biases in SDXL-generated images across six races, two genders, 32 professions, and eight attributes. We then propose a debiasing solution, called SDXL-Inc (where Inc stands for inclusive), and demonstrate its ability to outperform alternatives across various benchmarks. Additionally, using a measure of image similarity, we reveal that SDXL exhibits a high degree of racial homogenization, depicting individuals from certain racial backgrounds as being very similar to one another. We address this issue by proposing another solution, called SDXL-Div (where Div stands for diversity). Finally, using preregistered randomized controlled trials, we show that being exposed to inclusive AI-generated faces can reduce people's racial and gender biases, while exposure to non-inclusive AI-generated faces can increase these biases.

## 2 RELATED WORK

Compared to all the works discussed in this section, ours is the only one that examines, and addresses, the problem of racial homogenization in AI-generated faces, i.e., the depiction of individuals from a specific racial or ethnic group as too similar in appearance. We are also the first to conduct a randomized control trial to understand the impact of being exposed to inclusive and non-inclusive AI-generated faces, and whether the AI-label plays a role in this phenomenon. Next, we summarize relevant papers and discuss any additional differences that may exist between our work and theirs.

Bianchi et al. (2023) used a previous version of Stable Diffusion (v1-4) to examine biases across ten professions and three races. They classified the race and gender of any generated image as follows. First, for each of the five demographic categories considered (i.e., Male, Female, White, Black, Asian), they took the images that represent this category in the Chicago Face Dataset Ma et al. (2015), and fed those images to CLIP (the core representational component of Stable Diffusion) to generate vector representations. These were then averaged to obtain a single archetypal vector representation of the category (e.g., a single vector for Black). Any image can then be classified as $X$ if its vector representation is most closely aligned (in cosine distance) to the archetypal vector representation of $X$ (e.g., Black) compared to the alternative categories (e.g., Asian or White). The authors used this classifier to analyze 100 images per occupation, and found professional stereotypes that reinforce racial and gender disparities. Compared to our work, the authors did not consider certain racial groups, such as Indian, Latinx, and Middle Eastern. They also did not propose debiasing solutions.

Ghosh & Caliskan (2023) used a previous version of Stable Diffusion (v2.1) to analyze biases across three genders and 27 nationalities, but did not consider professions nor proposed debiasing solutions. Using CLIP's cosine similarity, they compared images generated using the prompt: *'a front-facing photo of a person'* against those generated using the prompt: *'a front-facing photo of $X$'* where $X$ is either a gender (i.e., a man, a woman, a nonbinary gender) or a nationality (e.g., a person from Brazil). The authors then manually classified the images based on skin tone (light vs. dark) and found that most generated images were light-skinned. Compared to our work, the authors did not consider professions (e.g., Doctor, Nurse, etc.) nor attributes (e.g., Beautiful, Intelligent, etc.) in their analysis. Moreover, in the context of race, the authors manually classified skin-color rather than proposing a classifier, and only provided qualitative rather than quantitative conclusions.

Wang et al. (2023) studied the association of pleasant and unpleasant attributes with certain concepts such as: European American and African American names, Light Skin and Dark Skin, Straight and Gay. In addition, they studied gender stereotypes across eight professions. Compared to our work, the authors did not propose any debiasing solutions. They also did not consider racial groups apart from European American (White) and African American (Black).

Friedrich et al. (2023) proposed a "user in control" solution, called Fair Diffusion, which utilizes a textual interface allowing users to instruct generative image models on fairness. Intuitively, this is made possible by extending classifier-free guidance Ho & Salimans (2022) with an additional fair guidance term, which depends on additional fairness instructions provided by the user. This approach requires no data filtering nor additional training. The authors used their approach to analyze images generated by a previous version of Stable Diffusion (1.5), focusing on professional stereotypes across genders. Compared to our work, the authors did not consider racial stereotypes, and did not consider a fully automated solution.

Zhang et al. (2023a) proposed a solution called ITI-GEN (inclusive Text-to-Image Generation), designed to generate images that are uniformly distributed across attributes of interest. One such attribute could be, e.g., "gender", in which case the images generated by ITI-GEN would be equally split between the various attribute categories, e.g., "male" and "female". More specifically, given an input prompt and some desired attribute(s), the model learns discriminative token embeddings representing each category of each attribute. It then injects these learned tokens after the original prompt, thereby synthesizing a set of prompts, each representing a unique combination of categories belonging to different attributes. Finally, this set is used to generate an equal number of images for any category combination. The authors used their model to analyze a previous version of Stable Diffusion (v1-4), focusing on 40 physical attributes (each consisting of two categories, one positive and one negative), along with gender, skin tone, and age.

Compared to our work, Zhang et al. focused on skin tone rather than race. Thus, unlike our analysis, theirs does not distinguish between, say, Asian and White individuals who happen to be equally light-skinned, or between Indian and Latinx individuals who happen to have similar skin tones. The authors also focus on physical attributes (e.g., Black hair, Mustache, etc.) as opposed to those describing various characteristics like the ones used in our analysis (e.g., Criminal, Intelligent, Parent, Poor, etc.). Additionally, their analysis of professional stereotypes focuses on four professions and 200 images per profession, while we focus on 32 professions and 10,000 images per profession. Finally, as discussed in Section D, their solution is unable to debias complex prompts, such as "*a person with green hair and eyeglasses*", or "*a person with the Eiffel tower*".

## 3 MATERIALS AND METHODS

### 3.1 DATA OVERVIEW

I) LAION-5B: This is the dataset that was used to train Stable Diffusion Schuhmann et al. (2022). We utilized a subset of this dataset, consisting of high-resolution images Beaumont (2021), to determine whether the biases in Stable Diffusion XL (SDXL) can be entirely attributed to the training data. We randomly selected 172,923 images from this subset, and kept those having one or more of the following keywords: face, person, child, woman, or man. We then cropped the images to retain only the face(s) appearing therein, and discarded any resulting face images that are smaller than $100 \times 100$ pixels. This filtering process left us with a final set of 88,714 face images.

II) FairFace: This is one of the largest public datasets of face images. For each image, the dataset specifies the race (Black, East Asian, Indian, Latinx, Middle Eastern, Southeast Asian, and White), and the gender (female, male) Karkkainen & Joo (2021). Moreover, the dataset is divided into two sets: 86,744 images for training and 10,954 for validation. We combined East and Southeast Asian into a single category: Asian. The resulting dataset was then used to train and validate our race and gender classifiers. For race, the number of images used for validation was: 1556 for Black; 2965 for Asian; 1516 for Indian; 1623 for Latinx or Hispanic; 1209 for Middle Eastern; and 2085 for White. As for gender, the number of images used for validation was: 5162 for female; and 5792 for male.

III) Flickr-Faces-HQ: This dataset consists of 70,000 high-resolution ($1024 \times 1024$) images of human faces crawled from Flickr with considerable variation in terms of age and race Karras & Hellsten (2023). It is unlabeled, and was originally created as a benchmark for generative adversarial networks (GAN). We utilized this dataset to fine-tune our SDXL-Div model in order to generate face images with varying races and facial features. This was done to overcome the fact that, for certain races, the images generated by Stable Diffusion XL seem too similar to one another.

IV) Stable Diffusion validation: This dataset consists of images that we have generated using SDXL. In particular, it consists of 10,000 images per race and 10,000 images per gender. For race, the prompt used to generate the images was: "*a photo of a X*", where $X \in$ {Asian, Black, Indian, Latino or Hispanic, Middle Eastern, White}. As for gender, the prompt used was: "*a photo of a X*", where $X \in$ {female, male}. This dataset was used to validate our race and gender classifiers.

V) SDXL-Inc fine-tuning: This dataset consists of Stable Diffusion-generated images with varying race, gender, and profession. The images were generated using the prompt: "*a photo of a X Y Z, looking at the camera, closeup headshot facing forward, ultra quality, sharp focus*", where $X \in$ {Asian, Black, Indian, Latino or Hispanic, Middle Eastern, White}; $Y \in$ {female, male}; and $Z$ is

one of the 21 professions listed in Appendix Table 1 under the "fine-tuning" category. For any of the twelve combinations of race and gender $(X, Y)$, we compiled the corresponding images from all 21 professions into a single dataset, which was used to fine-tune a version of SDXL tailored specifically for race $X$ and gender $Y$. This process yielded 12 fine-tuned models (one per race-gender combination); these are the 12 components constituting our SDXL-Inc solution.

VI) Profession: This dataset consists of SDXL-generated images depicting 32 professions. More specifically, we generated 10,000 images per profession using the prompt: "*a photo of Z, looking at the camera, closeup headshot facing forward, ultra quality, sharp focus*", where $Z$ is one of the 32 professions listed in Appendix Table 1. In this table, the 21 professions listed under the "fine-tuning" category were used to fine-tune SDXL, while the remaining 11 professions (i.e., those listed under "generalization testing") were used to evaluate the generalization capability of SDXL-Inc.

VII) Attribute: This dataset consists of Stable Diffusion-generated images depicting eight attributes. In particular, we generated 10,000 images per attribute using the prompt: "*a photo of a X, looking at the camera, closeup headshot facing forward, ultra quality, sharp focus*", where $X \in \{$Poor, Winner, Beautiful, Intelligent, Parent, Sibling, Terrorist, Criminal$\}$. This dataset was used to evaluate the generalization capability of SDXL-Inc.

### 3.2 METHODS

#### 3.2.1 STABLE DIFFUSION

We utilized SDXL due to its improved ability to generate human faces compared to its predecessors. We used the Hugging Face repository "stabilityai" and the model "stable-diffusion-xl-base-1.0" stabilityai (2023) to generate images to analyze racial and gender stereotypes in the context of various professions and attributes. We fine-tuned SDXL using LORA Hu et al. (2021), thereby creating our SDXL-Inc model. To this end, we created a dataset consisting of prompt-image pairs. Each such pair consisted of an image taken from the "SDXL-Inc fine-tuning dataset" (described above) along with the prompt: "*a photo of Z, looking at the camera, closeup headshot facing forward, ultra quality, sharp focus*", where $Z$ is the profession depicted in the image. Similarly, we fine-tuned SDXL using LORA in the process of creating our SDXL-Div model. To this end, we created a dataset consisting of prompt-image pairs. Each such pair consisted of an image taken from the "Flickr-Faces-HQ dataset" along with the prompt: "*a photo of X person, looking at the camera, closeup headshot facing forward, ultra quality, sharp focus*", where $X$ is the race depicted in the image.

**Hyper-parameters:** Image resolution = 1024; number of inference steps = 40; guidance scale = 5. When fine-tuning SDXL, we used: Image resolution = 1024; training batch size = 1; number of training epochs = 3; learning rate = $10^{-4}$; and mixed precision = fp16.

#### 3.2.2 PROPOSED CLASSIFICATION PIPELINE

Our classifier includes three stages: face detection, face embedding generation, and classification.

**I) Face detection:** This stage is carried out using a Multi-task Cascaded Convolutional Neural Network (MTCNN), which is a deep cascaded multitask framework that utilizes the inherent correlation between detection and alignment to improve performance Zhang et al. (2016). It leverages a cascaded architecture with three stages of deep convolutional networks to predict faces. We selected MTCNN due to its ability to balance high detection accuracy and run-time speed. We configured the detector to exclude boundary-boxes with confidence scores $\leq 0.9$.

**II) Face embedding generation:** This stage is carried out using a VGGFace ResNet-50 Convolutional Neural Networks (VGGFace ResNet-50 CNN) Cao et al. (2018). ResNet-50 CNN was trained on MS-Celeb-1M and VGGFace2, as well as the of the two. More specifically, MS-Celeb-1M has 10 million images depicting 100k different celebrities Guo et al. (2016). On the other hand, VGGFace is a large-scale face dataset with considerable variations in pose, age, illumination, race, and profession Cao et al. (2018), including 3.31 million images downloaded from Google image search. In our work, we utilized VGGFace ResNet-50 CNN, and removed the top layers to extract the embedding vector from the face images in the FairFace dataset described earlier. The FairFace RGB images were resized to 224x224 pixels before being fed to the CNN as input.

**III) Classification:** This stage is carried out using a Support Vector Machine (SVM) Hearst et al. (1998). We trained two SVM classifiers to predict the race and gender using the embedding vectors extracted from the FairFace images of the previous stage. The hyper-parameters used were: Regularization parameter C = 1; Kernel type = Radial Basis Function. This stage is repeated twice; once for race, and once for gender. However, the previous two stages are executed only once.

### 3.2.3 GPT-IN-THE-LOOP

In addition to SDXL-Inc, we experimented with an alternative debiasing technique. This prompt-regulating technique introduces an intermediary layer between the user (who provides the prompt) and SDXL. The layer uses GPT-4 to detect whether the user-provided prompt corresponds to the generation of an image depicting a person without a specific race and/or gender. GPT-4 would then inject a randomly-selected race (if race was not specified by the user) and/or a random gender (if gender was not specified) into the user-provided prompt before passing it on to SDXL. Given a user-provided prompt, $X$, the instruction given to GPT-4 is: "*For this text X: 1) select using one word ['yes','no'] if text includes any profession or a social media influencer 2) find the subject practicing the job. 3) select using one word ['yes','no'] if the text includes any country, nationality, or race or ethnicity. 4) select using one word ['female','male', 'unknown'] the subject's gender*"

## 4 RESULTS

### 4.1 CLASSIFYING GENDER AND RACE

To examine the biases in SDXL, we developed a race and gender classifier, composed of three stages: face detection, face embedding generation, and the classification stage (see Methods). To train and validate our classifier, we utilize FairFace Karkkainen & Joo (2021), which specifies the race (Black, East Asian, Indian, Latinx, Middle Eastern, Southeast Asian, and White), and gender (Female, Male) of each image[1]. We simplified FairFace's categorization by combining the East and Southeast Asian categories into a single one (Asian). We trained and evaluated our classifier on FairFace's training set and validation set, respectively. We benchmarked against several alternatives from the literature, namely: CLIP's zero-shot Radford et al. (2021), Google's FaceNet Schroff et al. (2015) + SVM Hearst et al. (1998), FairFace's ResNet-34 Karkkainen & Joo (2021), EfficientNet-B7 (tuning all layers) Tan & Le (2019), and Large Vision Transformer VIT (tuning all layers) Dosovitskiy et al. (2020). Section C summarizes the results, showing that our classifier consistently achieves state-of-the-art performance in terms of accuracy, precision, recall, and F1 score.

### 4.2 EXAMINING BIASES IN STABLE DIFFUSION

We examine the degree to which different races and genders appear in SDXL generated images. To this end, we generated 10,000 images using the following racial- and gender-neutral prompt: "*a photo of a person*". The distribution of the resulting images is summarized by the dashed bars in Figure 1. As can be seen, White is the most generated race (47% of images), followed by Black (33%). The remaining races are rarer in comparison, e.g., 3% are Asian, and 5% are Indian. As for gender, males appear more frequently (65%). These findings mirror what was reported by Ghosh & Caliskan (2023), showing that most images generated by Stable Diffusion (v2.1) representing the prompt "*a front-facing photo of a person*" depict light-skinned Western men.

One possible explanation behind these results could be that SDXL is merely reflecting the biases already present in the dataset on which it was trained, namely LAION-5B Schuhmann et al. (2022). To examine this possibility, we used a subset of LAION-5B Beaumont (2021) consisting of 88,714 images (see Material and Methods). The distribution of those images is summarized by the plain bars in Figure 1. As can be seen, images depicting White individuals are more frequent in LAION-5B than SDXL (63% vs. 47%), while those depicting other races are less frequent. As for gender, both male and female individuals appear in LAION-5B with equal probability. This indicates that SDXL contains biases that cannot be fully explained by the data it was trained on.

---

[1] These classes are widely studied in literature due to their availability from FairFace. It should be noted, however, that these classes do not capture the full spectrum of genders and races. Moreover, existing image classifiers (including our own) are meant to predict the perceived (rather than identified) gender and race.

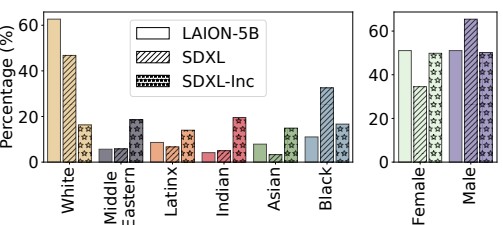

Figure 1: Examining gender and race distributions in LAION-5B, SDXL, and our SDXL-Inc in a sample of 88,714 images from the LAION-5B dataset, 10k images generated by SDXL, and 10k generated by SDXL-Inc. For the latter 20k images, we used the prompt: "*a photo of a person*".

Some users may consider the racial distribution of SDXL-generated images to be unsatisfactory, especially if it underrepresents certain groups compared to the society in which these users reside. One way to address this issue is to develop a solution that allows users to specify their desired distributions of race and gender. Perhaps the most intuitive target distribution is the one in which different groups are represented equally. With this in mind, we introduce such a debiasing solution, and test its ability to represent genders and races equally, although the same techniques can be used with any given target distribution. Specifically, we introduce a fine-tuned version of SDXL, which we call: "SDXL-Inc" (Inc stands for Inclusive).

Our model was created as follows. First, we identified 32 professions, and split them into 21 for fine-tuning, and 11 for testing (these professions are listed in Appendix Table 1). Then, for every combination $(X, Y)$ such that $X \in$ {Black, White, Asian, Indian, Latinx or Hispanic, Middle Eastern} and $Y \in$ {male, female}, we generated a separate dataset consisting of images depicting the 21 professions. The generation of these 12 datasets (6 races × 2 genders) was done using the prompt: "*a photo of X Y Z looking at the camera, closeup headshot facing forward, ultra quality, sharp focus*", where $Z$ denotes one of the 21 professions. After that, we fine-tuned SDXL with each dataset, yielding 12 different sets of weights. The fine-tuning was done using LORA (Low Rank Adaptation) Hu et al. (2021), which reduces the number of trainable parameters for the downstream task (see Materials and Methods). The basic idea of SDXL-Inc is to randomly select one of those 12 sets of weights based on the target distribution of interest (which is uniform in our experiments).

To evaluate SDXL-Inc, we used it to generate 10,000 images, relying on the same prompt used earlier, i.e., "*a photo of a person*". The distribution of the resulting images is summarized by the starred bars in Figure 1. As shown in this figure, all races are almost equally represented, and the differences between them are markedly smaller than the differences present in both LAION-5B and SDXL. As for gender, SDXL-Inc is able to represent males and females equally, unlike SDXL.

### 4.3 EXAMINING PROFESSIONAL STEREOTYPES IN STABLE DIFFUSION

We used SDXL to generate 320,000 images depicting 32 professions (10,000 per profession), using the prompt: "*a photo of a Z, looking at the camera, closeup headshot facing forward, ultra quality, sharp focus*", where $Z$ is the profession. Moreover, to specify the types of images that we want to avoid, we used the following negative prompt: "*cartoon, anime, 3d, painting, b&w, low quality*". For each profession, we used our classifier to examine the racial- and gender- composition of images.

Figure 2a depicts the results for 24 professions, while Figure 6 in the Appendix depicts the remaining eight professions. Numeric values below 15% are omitted to improve the visualization (see Table 2 in the Appendix for all values). As can be seen, White is the most frequently generated race in 21 out of the 24 professions. As for the remaining three, two of them are among the least prestige occupations, namely Cleaner and Security Guard Hofmann et al. (2024), and both are mostly represented by images depicting Black individuals. These findings confirm the findings of Bianchi et al. (2023), showing that prestigious, high-paying professions are often represented as White.

Figure 2b shows the gender distribution (see Table 2 in the Appendix for exact values). Males represent 90% of the images in 16 professions, including Doctor and Professor—two prestigious occupations Hofmann et al. (2024). Moreover, jobs in which women are more represented include Nurse and Secretary—two common stereotypes of women Friedrich et al. (2023); Wang et al. (2023).

### 4.4 DEBIASING STABLE DIFFUSION ACROSS PROFESSIONS AND ATTRIBUTES

Next, we examine stereotypes in terms of the following attributes: Winner, Beautiful, Intelligent, Parent, Sibling, Terrorist, Poor, and Criminal. The results are depicted in the upper row of Fig-

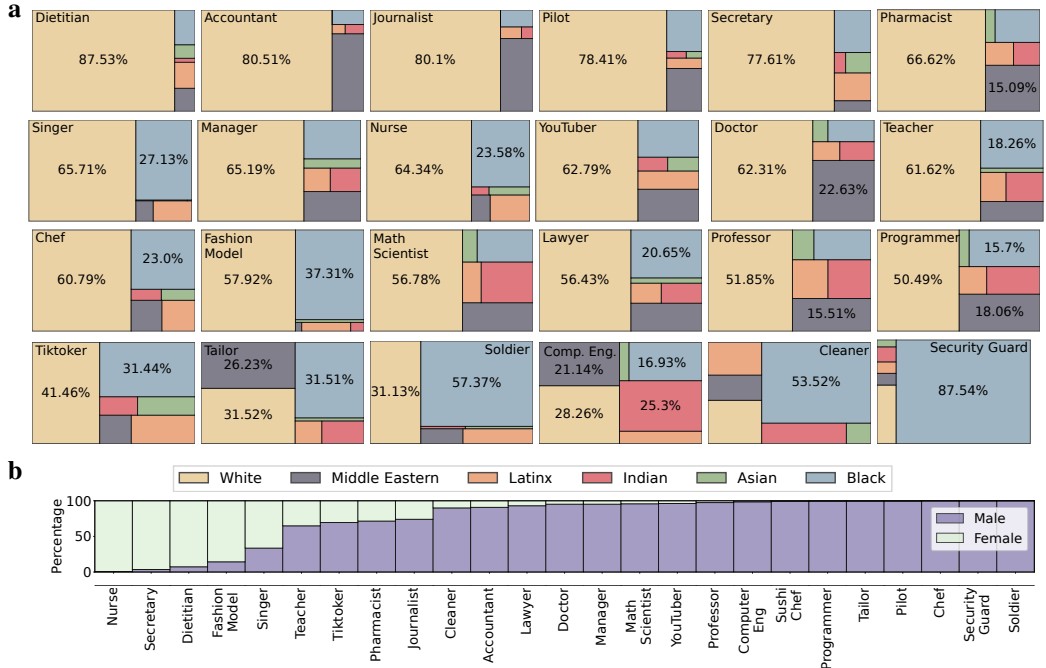

Figure 2: SDXL's professional stereotypes. SDXL was used to generate 10,000 images for each of the 25 professions. **a**, Racial distribution per profession. **b**, Gender distribution per profession.

ure 3a. White dominates the three attributes in our analysis that tend to be associated with success and attractiveness (Winner, Beautiful, and Intelligent). White also dominates the two family-related attributes (Parent and Sibling). In contrast, when it comes to Terrorism, Middle Eastern are the most common race, and none of the images depict a White individual, reinforcing existing stereotypes Kundnani (2014). Similarly, when it comes to crime and poverty, the majority of images depict Black individuals Quillian & Pager (2001).

Having established that SDXL exhibits biases in terms of attributes, we now evaluate SDXL-Inc's ability to address these biases. We repeated the same procedure used earlier with SDXL, but using our SDXL-Inc instead. The results for professions are depicted in the bottom row of Figure 3a. As shown, races are represented more uniformly compared to SDXL, as evidenced by the substantial reduction in standard deviation ($\sigma$). Importantly, none of the above eight attributes were used in the fine-tuning phase, and yet SDXL-Inc was able to significantly reduce the racial biases related to them. This indicates that SDXL-Inc can be generalized beyond the features it was fine-tuned on.

To further assess the generalizability of SDXL-Inc, we selected four of the professions used during the fine-tuning phase (Dietitian, Manager, Pharmacist, and Pilot), and four professions that were not used during that phase (Accountant, Journalist, Musician, and Firefighter). Figure 3b shows the racial distribution of images using SDXL (upper row) and SDXL-Inc (lower row). As can be seen, regardless of whether the profession is Black-dominated (Musician and Firefighter) or White-dominated, our solution is able to significantly reduce difference between races. Similar trends are observed when examining the remaining 24 professions (see Figure 7 and Table 4 in the Appendix).

Finally, we evaluate SDXL-Inc's ability to address gender biases across the aforementioned attributes and professions. As shown in Figure 3c, SDXL (solid bars) exhibits substantial biases, e.g., depicting the vast majority of intelligent individuals as male. In contrast, our solution consistently produces an equal, or near-equal, split between female and male (dashed bars). These improvements are reflected by the vast reduction in standard deviation, from 40.3 to just 2.7.

Additionally, we experimented with an alternative debiasing solution using a Large Language Model, namely GPT-4 OpenAI (2023), "in-the-loop" (see the Methods section). The results are depicted in Figure 8, and the exact values are listed in Table 5 in the Appendix. This solution is also capable of drastically reducing the race and gender biases exhibited by SDXL.

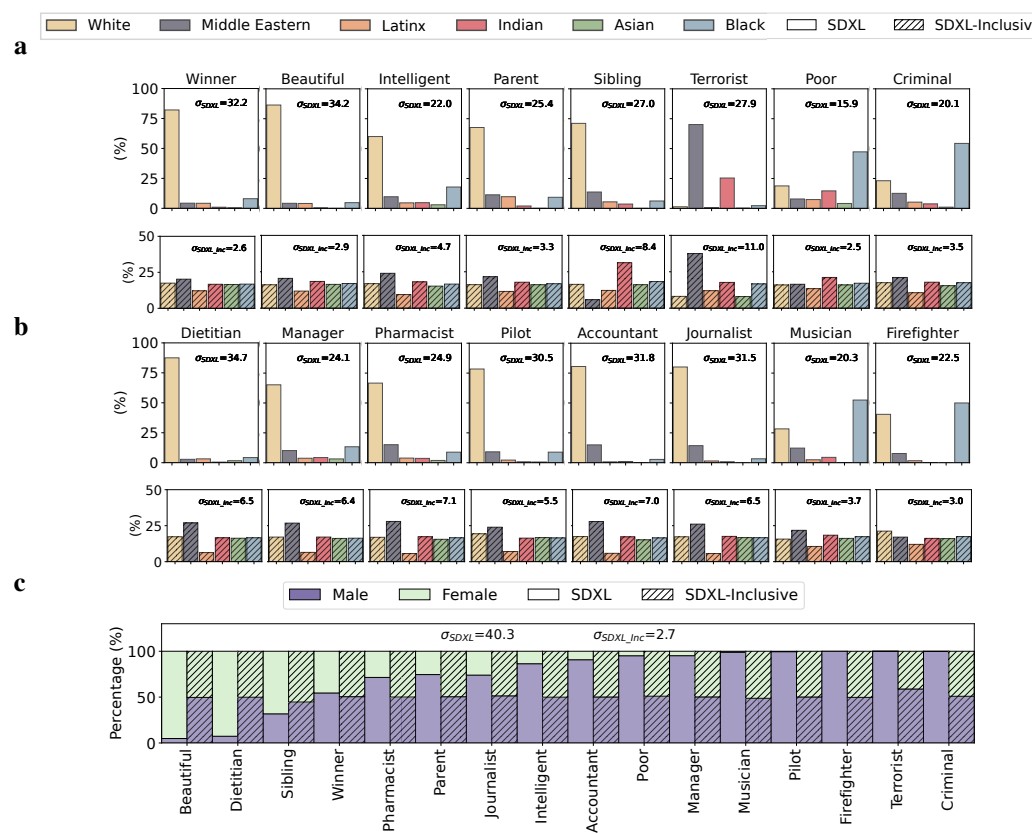

Figure 3: Results of SDXL-Inc. Given eight attributes and eight professions, both SDXL and SDXL-Inc were used to generate 10,000 images per profession and per attribute. **a**, Race distribution per attribute, with the upper row corresponding to SDXL, and the lower row corresponding to SDXL-Inc. **b**, The same as (**a**) but for professions instead of attributes. **c**, Gender distribution per profession and per attribute for SDXL and SDXL-Inc. The standard deviation(s) corresponding to each subplot is denoted by $\sigma$ followed by a subscript indicating the model.

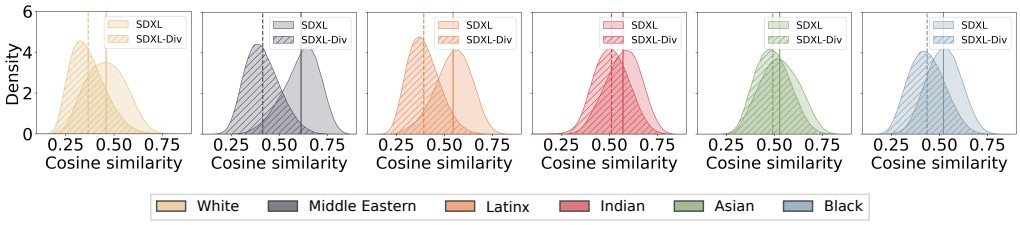

Figure 4: Cosine similarity of images generated using SDXL (plain) and SDXL-Div (dashed).

## 4.5 ADDRESSING RACIAL HOMOGENIZATION

We aimed to develop a version of SDXL capable of generating images with greater facial diversity per race compared to SDXL. To this end, we downloaded images from Flickr-Faces-HQ, which hosts high-resolution (1024×1024) images of human faces with "considerable variation in terms of age, race, and image background" Karras & Hellsten (2023). Since the dataset is unlabeled, we used our classifier to predict race (see Table 3 in the Appendix for the number of images per race). The resulting labelled dataset was then used to fine-tune SDXL, this was done using LORA Hu et al. (2021) to reduce the number of trainable parameters for the downstream task. We call the resulting model "SDXL-Div" (where Div is a shorthand for Diversity).

In our evaluation, each of the two models (SDXL and SDXL-Div) was used to generate $\sim$10,000 images per race. Our racial classifier was then used to obtain an embedding for each image. Finally, we computed the cosine similarity between every pair of images that have the same race and are generated by the same model, resulting in $\sim$50 million cosine similarity values per race per model. As shown in Figure 4, SDXL-Div is able to increase the facial diversity of SDXL, regardless of race. The greatest difference is observed for Middle Eastern (dropping the mean cosine similarity from 0.61 to 0.41) and Latinx (from 0.55 to 0.39); see Figures 20a and 20b in the Appendix for sample images of Middle Eastern individuals generated using SDXL and SDXL-Div, respectively.

## 4.6 USER STUDY

We ran four user studies to determine whether exposure to AI-generated faces can affect people's racial and gender biases. We estimated that to obtain a power of 0.8 to detect a medium effect size (Cohen's d) of 0.5 in a paired-sample comparison, a sample of 135 participants would be needed. We recruited participants on Prolific, with prescreening settings of US resident and English as native language. Additionally, we prohibited participation more than once in our experiment. These studies were preregistered at AsPredicted. All studies were approved by the Institutional Review Board under the category of Exempt or Expedited Research.

In each of the four studies, participants are presented with six AI-generated images, answer a few questions about each image, and then answer an overall question. Each study has four different conditions, depending on whether the images are inclusive or not, and whether participants are informed that the images are generated by AI or are produced by an artist. Next, we provide an overview of the four studies; see Section E in the Appendix for the exact wording and images used.

Study 1 examines racial bias. Given a profession $P \in \{\text{chef}, \text{dietitian}, \text{journalist}\}$, participants are presented with six images depicting $P$. For each image, they are asked to determine the age, perceived gender, and perceived race of the person depicted therein. After seeing all six images, participants answer the question $Q_1$: What percentage of $P$ in the US are White? Here, the non-inclusive images are generated using SDXL and they all depict white individuals, whereas the inclusive images are generated using SDXL-Inc, with each races depicted once. In both conditions, three images depict men, while the other three depict women. All images can be seen in Appendix Figure 12.

Study 2 examines gender bias, and is similar to Study 1 apart from two changes: (i) The professions are $\{\text{accountant}, \text{math scientist}, \text{and tailor}\}$; (ii) after seeing all six images, participants answer the question $Q_2$: What percentage of $P$ in the US are men? Here, the non-inclusive images are SDXL-generated, and all depict men, while inclusive images are generated by SDXL-Inc, representing men and women equally. In both conditions, races are represented equally (Appendix Figure 14).

Study 3 examines racial homogenization. Participants are presented with six images depicting Middle Eastern men. For each image, they are asked to describe the age, skin tone, and facial hair of the man featured therein. After seeing all six images, participants answer the question $Q_3$: What is your estimation of the percentage of Middle Eastern men who have beards? Here, the non-inclusive images are SDXL-generated, and all depict bearded men. In contrast, the inclusive ones are generated by SDXL-Div, and depict men with varying levels of facial hair (Appendix Figure 4).

Study 4 also examines racial homogenization, but focuses on women instead of men. Participants are presented with six images depicting Middle Eastern women, and are asked to describe the age, skin tone, and head cover of the woman featured in the image. Finally, participants answer the question $Q_4$: What is your estimation of the percentage of Middle Eastern women who wear headcovers? Here, the non-inclusive images are generated by SDXL, and depict women that all wear a head cover. On the other hand, the inclusive ones are generated by SDXL-Div and depict women, half of whom are wearing a head cover while the other half are showing their hair (Appendix Figure 18).

Additionally, for each question $Q_i : i \in \{1, 2, 3, 4\}$, we recruited 135 participants from Prolific to answer $Q_i$ without being presented with AI-generated images. Figures 5a to 5d summarize the responses to questions $Q1$ to $Q4$, respectively. For all questions, exposure to exposure to SDXL-generated images increase bias (apart from $Q_1$), while exposure to images generated by SDXL-Inc reduces bias, compared to the baseline in which participants are not exposed to AI-generated images. The figures also show no significant difference in participants' responses when the images are labelled as being produced by an artist rather than being AI-generated.

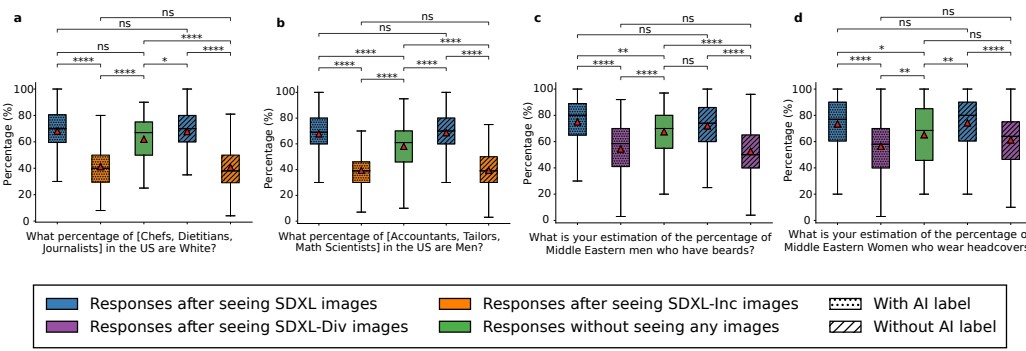

Figure 5: User study results. Subfigures **a** to **d** summarize the participants' responses in Studies 1 to 4, respectively. Boxes extend from the lower to upper quartile values, with a horizontal line at the median; whiskers extend to the most extreme values no further than 1.5 times the interquartile range from the box. P values are calculated using the t-test, unless one of the groups does not pass the Shapiro–Wilk test, in which case P values are calculated using the Mann-Whitney U test. $^*$p<0.05; $^{**}$p<0.01; $^{***}$p<0.001; $^{****}$p<0.0001; $ns$ = not significant).

## 5 DISCUSSION

We set out to examine the stereotypes and biases in SDXL, a text-to-image generator used daily by millions worldwide Fatunde & Tse (2022). To the end, we developed a classifier to predict the race and gender of any given face image, and demonstrated that it achieves state-of-the-art performance. Using this classifier, we showed that the vast majority of faces generated by SDXL are White males. Biases were also found when considering various attributes, e.g., associating beauty with femininity and intelligence with masculinity. Some of these biases are less severe in the dataset on which Stable Diffusion was trained, suggesting that certain biases are further exacerbated by the model itself.

Biased text-to-image models may contribute to the normalization of gender stereotypes, potentially shaping societal attitudes towards the roles and capabilities of women in various professions. For instance, as we have demonstrated, SDXL mostly depicts secretaries and nurses as women while depicting doctors and professors as men. Bearing in mind that millions worldwide are already using such models daily, addressing gender stereotypes in these models can be crucial. As suggested by Study 2, such stereotypes can be reduced by an inclusive model and can be exacerbated by a non-inclusive one, indicated the potential of AI in alleviating gender inequality.

Biased representations of gender and race may contribute to the creation of content that not only misrepresents certain groups but may also perpetuate discriminatory practices. This could be detrimental in the context of advertising, marketing, and media campaigns, where visuals hold substantial influence. For example, as our analysis has shown, Stable Diffusion associates low-income jobs such as Cleaner, Janitor, and Security Guard with Black people, while associating higher-prestige jobs such as Doctor, Lawyer, and Professor with White people. These findings reflect what has been reported by Bianchi et al. Bianchi et al. (2023), showing that several of the most prestigious, high-paying professions are represented by Stable Diffusion (v1-4) as White. Another example of bias is the association between crime and Black people, as well as the assassination between terrorism and Middle Easterners, which may reinforce existing biases against these racial groups Kundnani (2014); Quillian & Pager (2001). As we have seen in Study 1, whether or not the model is inclusive can affect people's perception of the racial distribution of certain professions, and this effect is likely to grow more pronounced as the use of AI-generated images becomes more widespread.

Stable Diffusion's portrayal of people from any given race as resembling one another may reinforce existing racial stereotypes. For instance, as we have demonstrated, Stable Diffusion mostly depicts Middle Eastern men as dark-skinned, bearded, and wearing a traditional headdress, and mostly depicts Middle Eastern women as dark-skinned, wearing a headscarf. Such oversimplified and generalized depictions of a particular racial group can be culturally insensitive, and may misrepresent the true diversity within that group. They may also lead to feelings of alienation, low self-esteem, and a sense of being misunderstood. As we have demonstrated in Studies 3 and 4, racial homogenization can be reduced using inclusive models, and can be exacerbated using non-inclusive ones.

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

## A SUPPLEMENTARY FIGURES

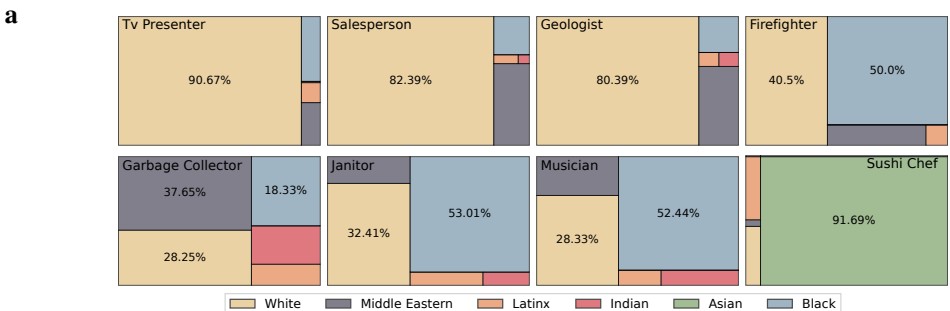

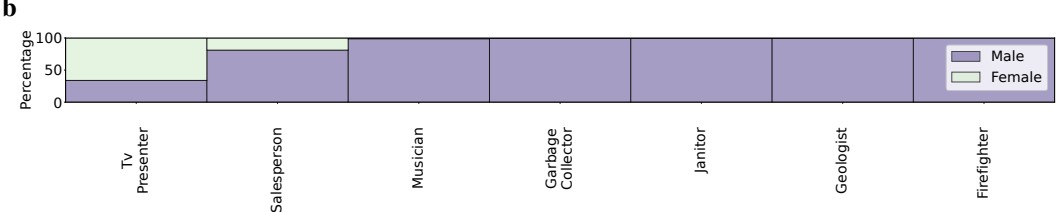

Figure 6: Professional stereotypes in Stable Diffusion XL (SDXL). Given the remaining eight professions that were not shown in the main article, SDXL was used to generate 10,000 images per profession. **a**, Racial distribution. **b**, Gender distribution.

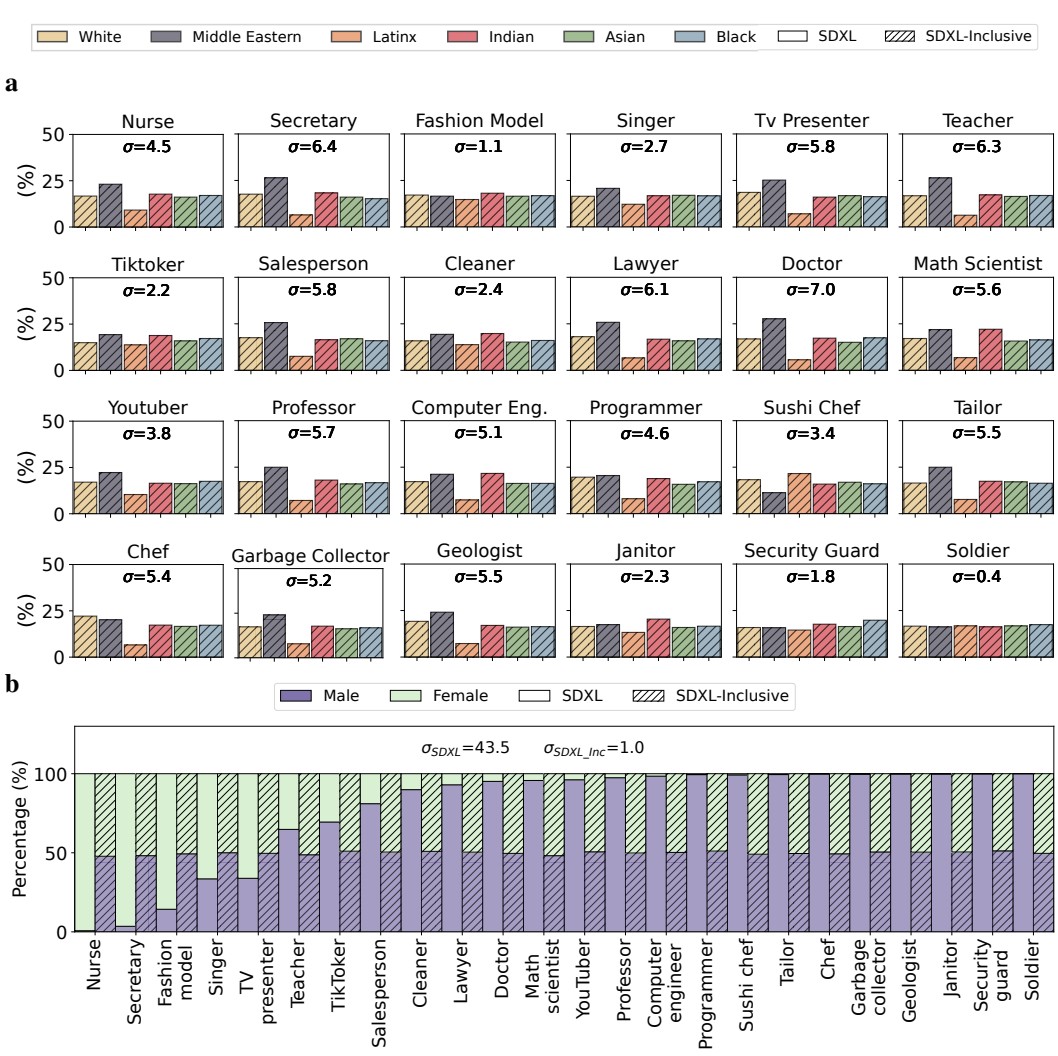

Figure 7: Results of SDXL-Inc. Given the remaining 24 professions that were not shown in the main article, SDXL-Inc was used to generate 10,000 images per profession. **a**, Race distribution. **b**, Gender distribution. The standard deviation(s) corresponding to each subplot is denoted by $\sigma$ followed by a subscript indicating the model.

a

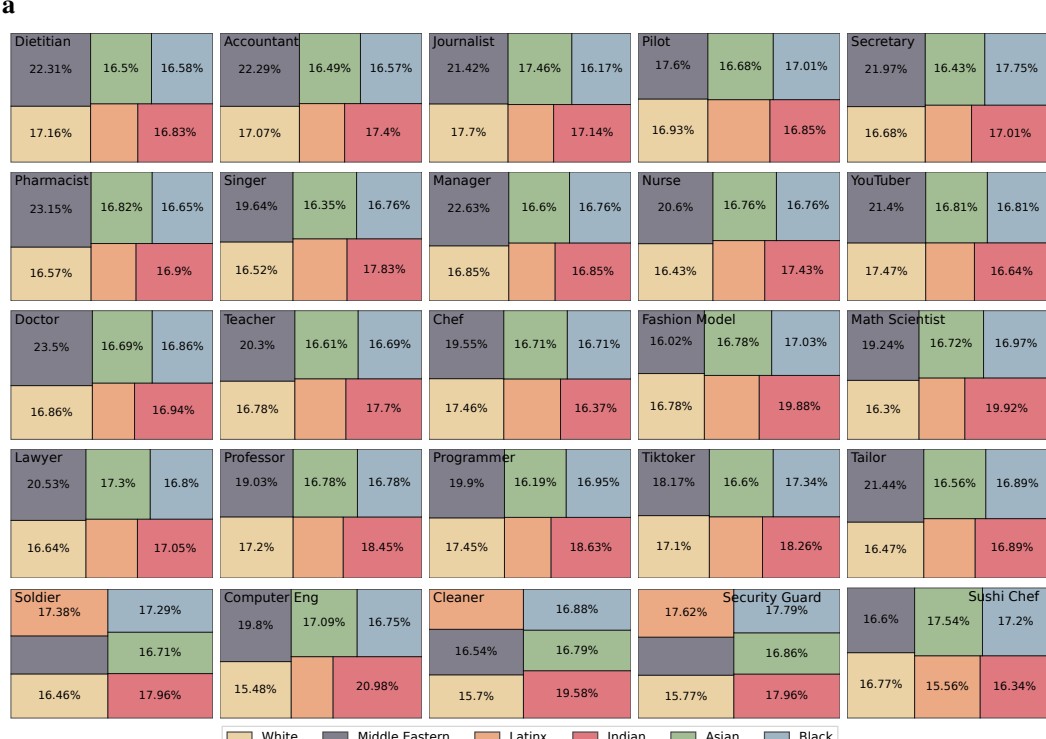

b

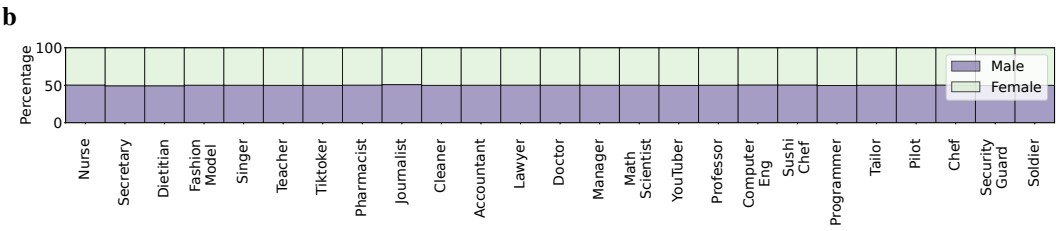

Figure 8: Results of our GPT-in-the-loop solution. For each profession, the race-, and gender-distribution of images generated by our GPT-in-the-loop solution. **a**, Race distribution. **b**, Gender distribution.

# B    SUPPLEMENTARY TABLES

| Profession | Usage |
|---|---|
| Accountant | Generalization testing |
| Chef | Fine-tuning |
| Cleaner | Fine-tuning |
| Computer Engineer | Fine-tuning |
| Dietitian | Fine-tuning |
| Doctor | Fine-tuning |
| Fashion Model | Fine-tuning |
| Firefighter | Generalization testing |
| Garbage Collector | Generalization testing |
| Geologist | Generalization testing |
| Janitor | Generalization testing |
| Journalist | Generalization testing |
| Lawyer | Fine-tuning |
| Manager | Fine-tuning |
| Mathematics Scientist | Fine-tuning |
| Musician | Generalization testing |
| Nurse | Fine-tuning |
| Pharmacist | Fine-tuning |
| Pilot | Fine-tuning |
| Professor | Fine-tuning |
| Programmer | Fine-tuning |
| Sales person | Generalization testing |
| Secretary | Fine-tuning |
| Security Guard | Fine-tuning |
| Singer | Generalization testing |
| Soldier | Fine-tuning |
| Sushi Chef | Fine-tuning |
| Tailor | Fine-tuning |
| Teacher | Fine-tuning |
| TikToker | Generalization testing |
| TV Presenter | Generalization testing |
| YouTuber | Fine-tuning |

Table 1: Professions and how they were used in our study.

| Profession | White | M.E. | Latinx | Indian | Asian | Black | Female |
|---|---|---|---|---|---|---|---|
| Accountant | 80.51 | 14.93 | 0.73 | 1.05 | 0.01 | 2.78 | 9.27 |
| Chef | 60.79 | 5.75 | 6.18 | 2.02 | 2.25 | 23.0 | 0.18 |
| Cleaner | 14.08 | 8.21 | 10.66 | 10.5 | 3.03 | 53.52 | 10.09 |
| Computer Engineer | 28.26 | 21.14 | 6.16 | 25.3 | 2.21 | 16.93 | 1.55 |
| Dietitian | 87.53 | 2.82 | 3.21 | 0.49 | 1.67 | 4.28 | 92.69 |
| Doctor | 62.31 | 22.63 | 3.08 | 3.92 | 1.99 | 6.08 | 4.85 |
| Fashion Model | 57.92 | 0.33 | 2.58 | 0.7 | 1.16 | 37.31 | 85.71 |
| Firefighter | 40.5 | 7.73 | 1.69 | 0.08 | 0.0 | 50.0 | 0.0 |
| Garbage Collector | 28.25 | 37.65 | 5.59 | 10.18 | 0.0 | 18.33 | 0.29 |
| Geologist | 80.39 | 12.01 | 1.08 | 1.05 | 0.0 | 5.47 | 0.17 |
| Janitor | 32.41 | 8.51 | 3.71 | 2.36 | 0.0 | 53.01 | 0.18 |
| Journalist | 80.1 | 14.29 | 1.49 | 0.81 | 0.0 | 3.31 | 26.03 |
| Lawyer | 56.43 | 12.0 | 3.7 | 4.94 | 2.28 | 20.65 | 7.03 |
| Manager | 65.19 | 10.18 | 3.76 | 4.32 | 3.19 | 13.36 | 4.85 |
| Math Scientist | 56.78 | 12.11 | 4.62 | 12.79 | 2.86 | 10.84 | 4.28 |
| Musician | 28.33 | 12.24 | 2.48 | 4.5 | 0.01 | 52.44 | 1.19 |
| Nurse | 64.34 | 2.97 | 6.26 | 0.86 | 1.99 | 23.58 | 99.29 |
| Pharmacist | 66.62 | 15.09 | 3.89 | 3.64 | 1.92 | 8.84 | 28.54 |
| Pilot | 78.41 | 9.12 | 2.22 | 0.78 | 0.62 | 8.86 | 0.48 |
| Professor | 51.85 | 15.51 | 8.37 | 10.1 | 3.88 | 10.28 | 2.54 |
| Programmer | 50.49 | 18.06 | 4.61 | 8.91 | 2.22 | 15.7 | 0.59 |
| Salesperson | 82.39 | 11.17 | 0.84 | 0.38 | 0.0 | 5.22 | 18.97 |
| Secretary | 77.61 | 2.32 | 6.16 | 1.39 | 3.12 | 9.4 | 96.48 |
| Security Guard | 7.03 | 1.41 | 1.38 | 1.81 | 0.83 | 87.54 | 0.1 |
| Singer | 65.71 | 2.15 | 4.75 | 0.24 | 0.02 | 27.13 | 66.49 |
| Soldier | 31.13 | 3.74 | 6.2 | 0.62 | 0.94 | 57.37 | 0.01 |
| Sushi Chef | 3.31 | 0.37 | 3.55 | 0.08 | 91.69 | 1.01 | 0.79 |
| Tailor | 31.52 | 26.23 | 3.66 | 5.78 | 1.29 | 31.51 | 0.55 |
| Teacher | 61.62 | 7.44 | 4.5 | 6.56 | 1.62 | 18.26 | 35.21 |
| TikToker | 41.46 | 5.44 | 10.97 | 4.25 | 6.44 | 31.44 | 30.56 |
| TV Presenter | 90.67 | 3.09 | 1.45 | 0.09 | 0.0 | 4.7 | 66.14 |
| YouTuber | 62.79 | 11.72 | 6.73 | 2.49 | 2.59 | 13.68 | 3.84 |

Table 2: SDXL's race and gender distribution per profession.

| | Black | Asian | Indian | Latinx | Middle east | White |
|---|---|---|---|---|---|---|
| Number of images | 2151 | 3255 | 2193 | 2731 | 1551 | 4071 |

Table 3: Number of images per race for the Flickr-Face-HQ dataset, as determined by our race classifier.

| Profession | White | M.E. | Latinx | Indian | Asian | Black | Female |
|---|---|---|---|---|---|---|---|
| Accountant | 17.43 | 27.9 | 5.82 | 17.21 | 15.15 | 16.49 | 49.86 |
| Chef | 22.14 | 20.16 | 6.56 | 17.3 | 16.6 | 17.24 | 50.64 |
| Cleaner | 15.84 | 19.35 | 13.79 | 19.78 | 15.2 | 16.04 | 49.09 |
| Computer Engineer | 17.22 | 21.2 | 7.37 | 21.65 | 16.27 | 16.29 | 49.77 |
| Dietitian | 17.3 | 26.87 | 6.31 | 16.61 | 16.31 | 16.6 | 50.13 |
| Doctor | 16.85 | 27.68 | 5.66 | 17.3 | 15.07 | 17.45 | 50.37 |
| Fashion Model | 17.16 | 16.58 | 14.76 | 18.15 | 16.54 | 16.81 | 50.65 |
| Firefighter | 21.23 | 17.09 | 12.03 | 16.2 | 16.02 | 17.43 | 50.29 |
| Garbage Collector | 17.33 | 24.01 | 7.84 | 17.76 | 16.25 | 16.8 | 49.47 |
| Geologist | 19.2 | 24.12 | 7.27 | 17.01 | 16.09 | 16.31 | 49.58 |
| Janitor | 16.43 | 17.43 | 13.25 | 20.41 | 15.9 | 16.59 | 49.51 |
| Journalist | 17.25 | 26.12 | 5.67 | 17.57 | 16.73 | 16.66 | 48.62 |
| Lawyer | 18.08 | 25.82 | 6.64 | 16.75 | 15.84 | 16.87 | 49.59 |
| Manager | 17.1 | 26.82 | 6.47 | 17.13 | 16.15 | 16.33 | 49.78 |
| Math Scientist | 17.1 | 21.92 | 6.81 | 22.08 | 15.76 | 16.33 | 51.83 |
| Musician | 15.7 | 21.79 | 10.58 | 18.41 | 16.18 | 17.35 | 51.29 |
| Nurse | 16.73 | 23.13 | 9.17 | 17.78 | 16.16 | 17.04 | 52.2 |
| Pharmacist | 16.93 | 27.94 | 5.67 | 17.31 | 15.55 | 16.6 | 49.92 |
| Pilot | 19.36 | 23.97 | 7.07 | 16.38 | 16.74 | 16.47 | 49.86 |
| Professor | 17.22 | 24.98 | 7.11 | 18.07 | 15.96 | 16.65 | 50.14 |
| Programmer | 19.63 | 20.51 | 8.04 | 18.88 | 15.8 | 17.14 | 48.92 |
| Salesperson | 17.54 | 25.66 | 7.53 | 16.43 | 16.96 | 15.9 | 49.55 |
| Secretary | 17.63 | 26.42 | 6.44 | 18.27 | 16.05 | 15.19 | 51.82 |
| Security Guard | 15.87 | 15.75 | 14.51 | 17.66 | 16.41 | 19.8 | 48.78 |
| Singer | 16.51 | 20.77 | 12.21 | 16.77 | 17.01 | 16.73 | 49.96 |
| Soldier | 16.62 | 16.19 | 16.76 | 16.26 | 16.73 | 17.42 | 50.35 |
| Sushi chef | 18.27 | 11.28 | 21.58 | 15.92 | 16.91 | 16.05 | 50.89 |
| Tailor | 16.48 | 24.98 | 7.65 | 17.43 | 17.09 | 16.37 | 50.45 |
| Teacher | 16.76 | 26.36 | 6.32 | 17.23 | 16.41 | 16.92 | 51.25 |
| TikToker | 14.96 | 19.31 | 13.77 | 18.84 | 15.91 | 17.22 | 48.98 |
| TV Presenter | 18.58 | 25.21 | 7.06 | 16.06 | 16.82 | 16.27 | 50.35 |
| YouTuber | 17.07 | 22.26 | 10.39 | 16.49 | 16.26 | 17.52 | 49.37 |

Table 4: SDXL-Inc's race and gender distribution per profession.

| Profession | White | M.E. | Latinx | Indian | Asian | Black | Female |
|---|---|---|---|---|---|---|---|
| Accountant | 17.07 | 22.29 | 10.19 | 17.4 | 16.49 | 16.57 | 50.04 |
| Chef | 17.46 | 19.55 | 13.2 | 16.37 | 16.71 | 16.71 | 49.79 |
| Cleaner | 15.7 | 16.54 | 14.51 | 19.58 | 16.79 | 16.88 | 50.21 |
| Comp. Engineer | 15.48 | 19.8 | 9.9 | 20.98 | 17.09 | 16.75 | 49.66 |
| Dietitian | 17.16 | 22.31 | 10.61 | 16.83 | 16.5 | 16.58 | 50.75 |
| Doctor | 16.86 | 23.5 | 9.14 | 16.94 | 16.69 | 16.86 | 50.0 |
| Fashion Model | 16.78 | 16.02 | 13.51 | 19.88 | 16.78 | 17.03 | 50.0 |
| Journalist | 17.7 | 21.42 | 10.11 | 17.14 | 17.46 | 16.17 | 49.15 |
| Lawyer | 16.64 | 20.53 | 11.67 | 17.05 | 17.3 | 16.8 | 49.92 |
| Manager | 16.85 | 22.63 | 10.31 | 16.85 | 16.6 | 16.76 | 50.04 |
| Math Scientist | 16.3 | 19.24 | 10.84 | 19.92 | 16.72 | 16.97 | 50.0 |
| Nurse | 16.43 | 20.6 | 12.01 | 17.43 | 16.76 | 16.76 | 49.79 |
| Pharmacist | 16.57 | 23.15 | 9.91 | 16.9 | 16.82 | 16.65 | 49.88 |
| Pilot | 16.93 | 17.6 | 14.93 | 16.85 | 16.68 | 17.01 | 50.04 |
| Professor | 17.2 | 19.03 | 11.77 | 18.45 | 16.78 | 16.78 | 50.0 |
| Programmer | 17.45 | 19.9 | 10.88 | 18.63 | 16.19 | 16.95 | 50.34 |
| Secretary | 16.68 | 21.97 | 10.16 | 17.01 | 16.43 | 17.75 | 50.78 |
| Security Guard | 15.77 | 14.0 | 17.62 | 17.96 | 16.86 | 17.79 | 50.42 |
| Singer | 16.52 | 19.64 | 12.9 | 17.83 | 16.35 | 16.76 | 50.04 |
| Soldier | 16.46 | 14.2 | 17.38 | 17.96 | 16.71 | 17.29 | 50.13 |
| Sushi Chef | 16.77 | 16.6 | 15.56 | 16.34 | 17.54 | 17.2 | 49.7 |
| Tailor | 16.47 | 21.44 | 11.75 | 16.89 | 16.56 | 16.89 | 50.08 |
| Teacher | 16.78 | 20.3 | 11.91 | 17.7 | 16.61 | 16.69 | 50.17 |
| TikToker | 17.1 | 18.17 | 12.53 | 18.26 | 16.6 | 17.34 | 50.21 |
| YouTuber | 17.47 | 21.4 | 10.87 | 16.64 | 16.81 | 16.81 | 50.25 |

Table 5: GPT-in-the-loop SDXL race and gender distribution per profession.

## C  OUR RACIAL AND GENDER CLASSIFIER

Figure 9a depicts the confusion matrices corresponding to race and gender, along with sample images representing each cell. As can be seen, Asian is the easiest race to predict, followed by Black. In contrast, the hardest races to predict are Latinx and Middle Eastern. On the other hand, there is no difference in performance when it comes to predicting male vs. female images.

We reproduced the same confusion matrices, but using images generated by Stable Diffusion XL (SDXL) which has been shown to outperform previous versions of Stable Diffusion Podell et al. (2023). More specifically, we generated 10,000 images per race using the following prompt: "*a photo of X person, looking at the camera, closeup headshot facing forward, ultra quality, sharp focus*", where $X \in \{$a Black, a White, an Asian, an Indian, a Latinx or Hispanic, a Middle Eastern$\}$. As for gender, we generated 10,000 images per gender using the same prompt as before, but with $X \in \{$male, female$\}$. The results of this evaluation are depicted in the first two rows of Table 8, showing that our classifier works better with images from SDXL than those from FairFace, regardless of the performance measure. This indicates that our classifier is particularly suited to examine the racial-, and gender-composition of images generated by SDXL.

The confusion matrices depicted in Figure 9b reveal that all races are significantly easier to predict in the SDXL dataset compared to its FairFace counterpart, apart from Latinx which is significantly harder (see how the numbers along the diagonal are greater in Figure 9b than in Figure 9a, except for Latinx). One possible explanation could be that the dataset used to train SDXL contains several images that are labelled as Latinx but are highly similar to Indian and Middle Eastern individuals. In terms of gender, our classifier achieves a perfect score for both male and female.

Table 6: A comparison between our **race classifier** and other literature alternatives, using FairFace's validation set and FairFace's seven-race classification. Bold font highlights the highest score(s).

|  | Accuracy | Precision | Recall | F1 score |
|---|---|---|---|---|
| Our classifier | **73%** | **72%** | **72%** | **72%** |
| CLIP's zero-shot classifier | 64% | 67% | 65% | 65% |
| Google's FaceNet + SVM | 69% | 69% | 68% | 68% |
| FairFace's (ResNet34) classifier | 72% | **72%** | 71% | **72%** |
| EfficientNet-B7 (tuning all layers) | 70% | 70% | 70% | 70% |
| Large Vision Transformer VIT (tuning all layers) | 71% | 72% | 70% | 71% |

Table 7: A comparison between our **gender classifier** and other alternatives from the literature, using FairFace's validation set. Bold font highlights the highest score(s).

|  | Accuracy | Precision | Recall | F1 score |
|---|---|---|---|---|
| Our classifier | **94%** | **94%** | **94%** | **94%** |
| FairFace's (ResNet34) classifier | **94%** | **94%** | **94%** | **94%** |
| CLIP's zero-shot classifier | **94%** | **94%** | **94%** | **94%** |

Table 8: Evaluating our classifier's ability to predict race and gender using SDXL images and our six racial groups. Bold font highlights the highest score.

|  | Accuracy | | Precision | | Recall | | F1 score | |
|---|---|---|---|---|---|---|---|---|
|  | FairFace | SDXL | FairFace | SDXL | FairFace | SDXL | FairFace | SDXL |
| Racial | 78% | **88%** | 76% | **90%** | 75% | **88%** | 76% | **87%** |
| Gender | 94% | **100%** | 94% | **100%** | 94% | **100%** | 94% | **100%** |

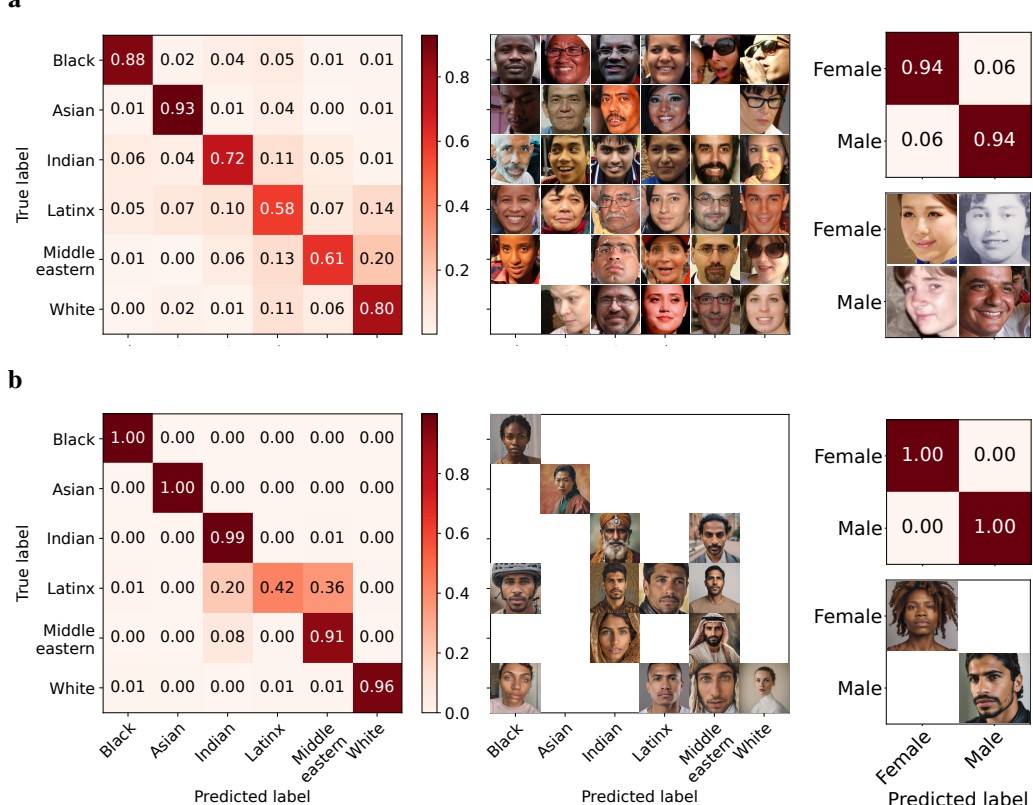

Figure 9: Evaluating our classifier. **a**, Confusion matrices of race and gender prediction based on the FairFace validation set, along with sample images representing different cells. **b**, Similar to (**a**), but based on images generated by Stable Diffusion XL (SDXL).

## D SDXL-INC VS ITI-GEN

In this section, we compare our solution (SDXL-Inc) to another recently proposed alternative, namely ITI-GEN Zhang et al. (2023a).

### D.1 RETRAINING ITI-GEN

We needed to benchmark SDXL-Inc against ITI-GEN Zhang et al. (2023a;b) in terms of how well it can debias Stable Diffusion. To this end, we retrained ITI-GEN with the six races (Asian, Black, Indian, Latino or Hispanic, Middle Eastern, and White) and the two genders (female and male) that SDXL-Inc was trained on. The training data used for this purpose was created as follows: First, we inferred the race and gender of each image in the Flickr-Faces-HQ dataset Karras & Hellsten (2023) using our classifier. Then, we curated two training-sets of images: one for gender and one for race. More specifically, for the gender training set, we randomly selected 50 images (from the aforementioned labeled Flickr dataset) for each gender. On the other hand, for the race training set, we randomly selected 25 images for each race.

To validate our resultant ITI-GEN model, we generated 1,200 images (100 per gender-race combination) using the prompt "*a photo of a person*". Figure 10 depicts the race and gender distribution across the 1,200 generated images. The results demonstrate that our trained version of ITI-GEN is capable of generating equal representation for gender, and nearly equal representation for race (apart from Latinx, due to their facial similarity to both White and Middle Eastern).

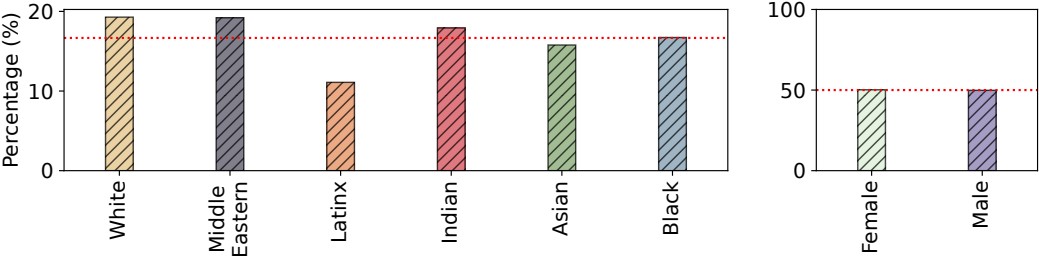

Figure 10: Race and Gender distribution obtained by the retrained ITI-GEN model. Comparing the representation of different races and genders in a sample of 1200 images generated by the retrained ITI-GEN model using the prompt: *"a photo of a person"*.

### D.2 PERFORMANCE COMPARISON

In ITI-GEN official GitHub repository Zhang et al. (2023b), the authors imply that their solution may struggle with certain generalizations. More specifically, they state that the training prompt (e.g., a headshot of a person) and the inference prompt (e.g., a headshot of a doctor) "should not differ a lot". To compare the degree to which both models (ITI-GEN and SDXL-Inc) can be generalized, we used each of them to generate 1,200 images for each of the following five prompts: 1) a headshot of a person with the Eiffel Tower in the background; 2) a headshot of a mechanic with a guitar; 3) a headshot of a skillful trainer with a pet tiger; 4) a headshot of a happy and healthy family; and 5) a headshot of a person with green hair and eyeglasses.

Figure 11 depicts the race and gender distribution of SDXL, SDXL-Inc, and ITI-GEN for each of the five prompts. As can be seen in the left column, SDXL is extremely biased, depicting the vast majority of images as White in all prompts. As for ITI-GEN (middle column, dashed bars), it manages to reduce a substantial amount of the bias compared to SDXL. Nevertheless, the percentage of images depicting White individuals remains ≥40% in four out of the five prompts. Our SDXL-Inc (middle column, starred bars) yields a more even distribution of images across races compared to ITI-GEN; see how the standard deviations for our solution are smaller than those of ITI-GEN in all five prompts. As for gender (right column), we can see that SDXL is the most biased, ITI-GEN reduces a considerable amount of bias, and SDXL-Inc further reduces bias compared to ITI-GEN across all prompts; see the corresponding standard deviations.

One reason behind these observed differences in performance could be the fact that the models occasionally intend to generate images of a certain race, but accidentally end up generating images of a different race (note that both SDXL-Inc and ITI-GEN contain steps designed to generate an image of certain races). If these steps work perfectly well, we would expect the models to achieve a perfect score in terms of accuracy, recall, precision, and F1 score. To test this possibility, we used out classifier to label the 1,200 images generated by each solution (200 per race) for each of the five prompts. The result of this analysis is summarized in Table 9. As can be seen, SDXL-Inc outperforms ITI-GEN across all metrics in all five prompts. This suggests that the observed difference in performance between the two solutions could be (at least partially) explained by the fact that the error rate in generating the intended race(s) is higher in ITI-GEN than in SDXL-Inc.

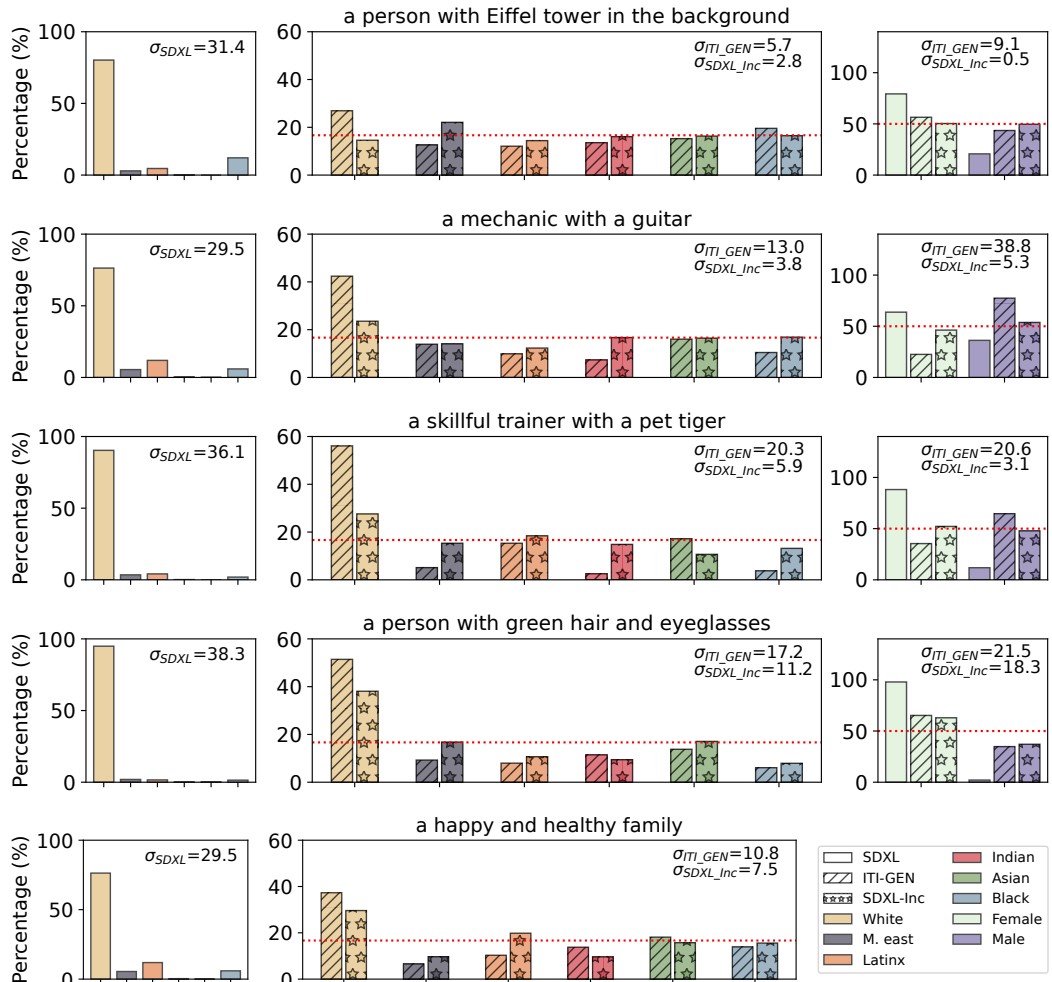

Figure 11: SDXL vs. ITI-GEN vs. SDXL-Inc. Comparing the distribution of race and gender across the three models, given five prompts. Each row corresponds to a different prompt. The standard deviation(s) corresponding to each subplot is denoted by $\sigma$ followed by a subscript indicating the model. Gender results are omitted from the bottom row since the generated images depict families rather than individuals.

Table 9: Comparing SDXL-Inc to ITI-GEN in terms of accuracy, recall, precision, and F1 score. For each performance measure, the highest score is highlighted in bold.

| Prompt | Accuracy | | Recall | | Precision | | F1 score | |
| --- | --- | --- | --- | --- | --- | --- | --- | --- |
| | SDXL-Inc | ITI-GEN | SDXL-Inc | ITI-GEN | SDXL-Inc | ITI-GEN | SDXL-Inc | ITI-GEN |
| a person with Eiffel tower in the background | **87** | 70 | **87** | 70 | **89** | 71 | **87** | 67 |
| a mechanic with a guitar | **86** | 39 | **86** | 39 | **88** | 48 | **86** | 37 |
| a skillful trainer with a pet tiger | **77** | 37 | **77** | 37 | **83** | 47 | **77** | 32 |
| a happy and healthy family | **70** | 66 | **70** | 66 | **75** | 68 | **70** | 64 |
| a person with green hair and eyeglasses | **60** | 40 | **60** | 40 | **72** | 55 | **60** | 38 |
| **Average** | **76** | 50.4 | **76** | 50.4 | **81.4** | 57.8 | **76** | 47.6 |

## E    USER STUDY IMAGES AND SAMPLE SCREENSHOTS

a

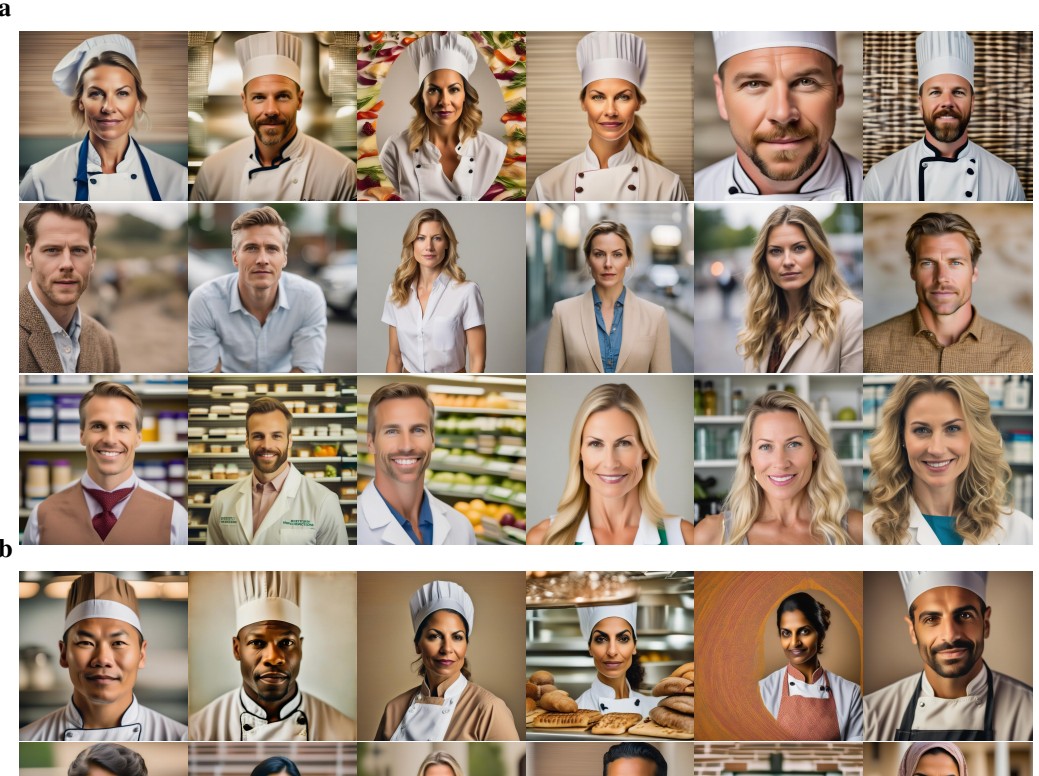

b

Figure 12: Study 1 images (racial). **a**, SDXL. **b**, SDXL-Inclusive.

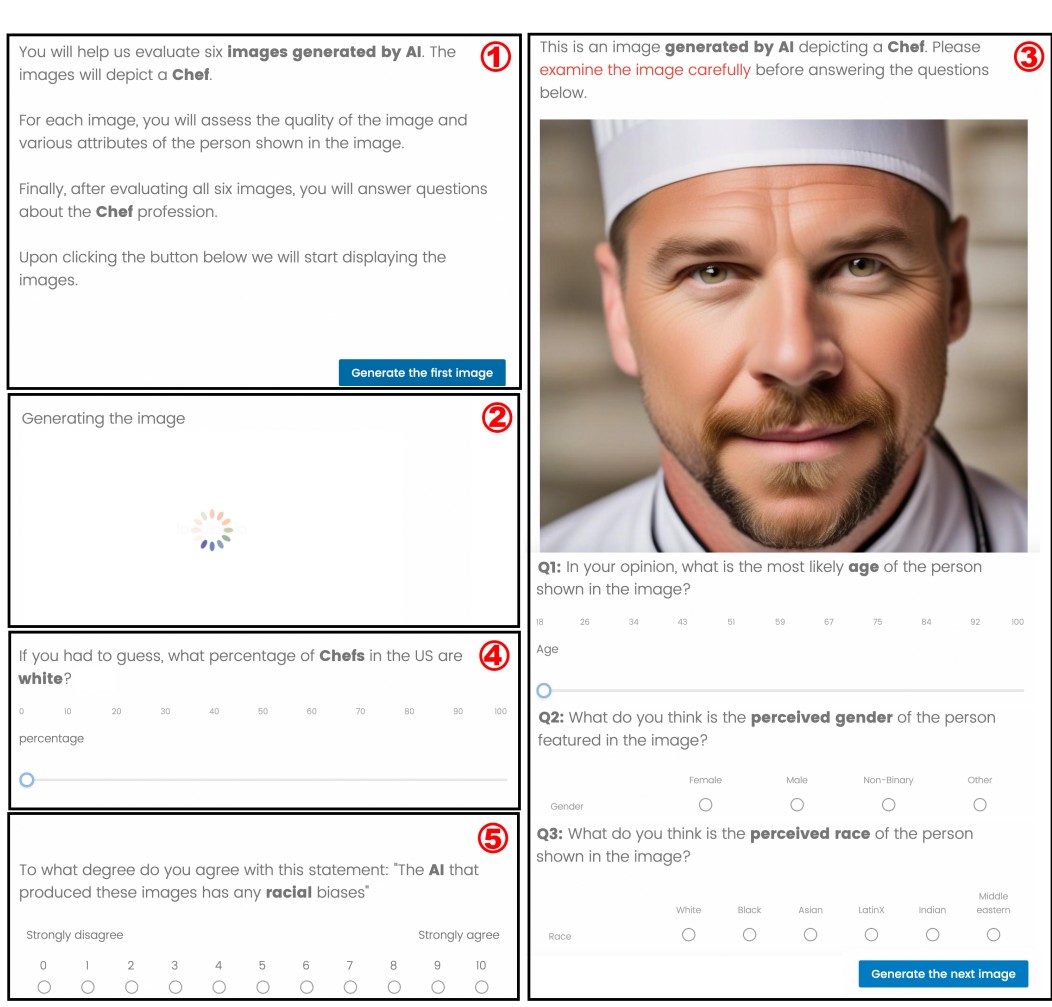

Figure 13: Study 1 sample screenshots. (1) welcome screen, (2) loading screen, (3) per image questions, and (4)/(5) Final questions after seeing all six images.

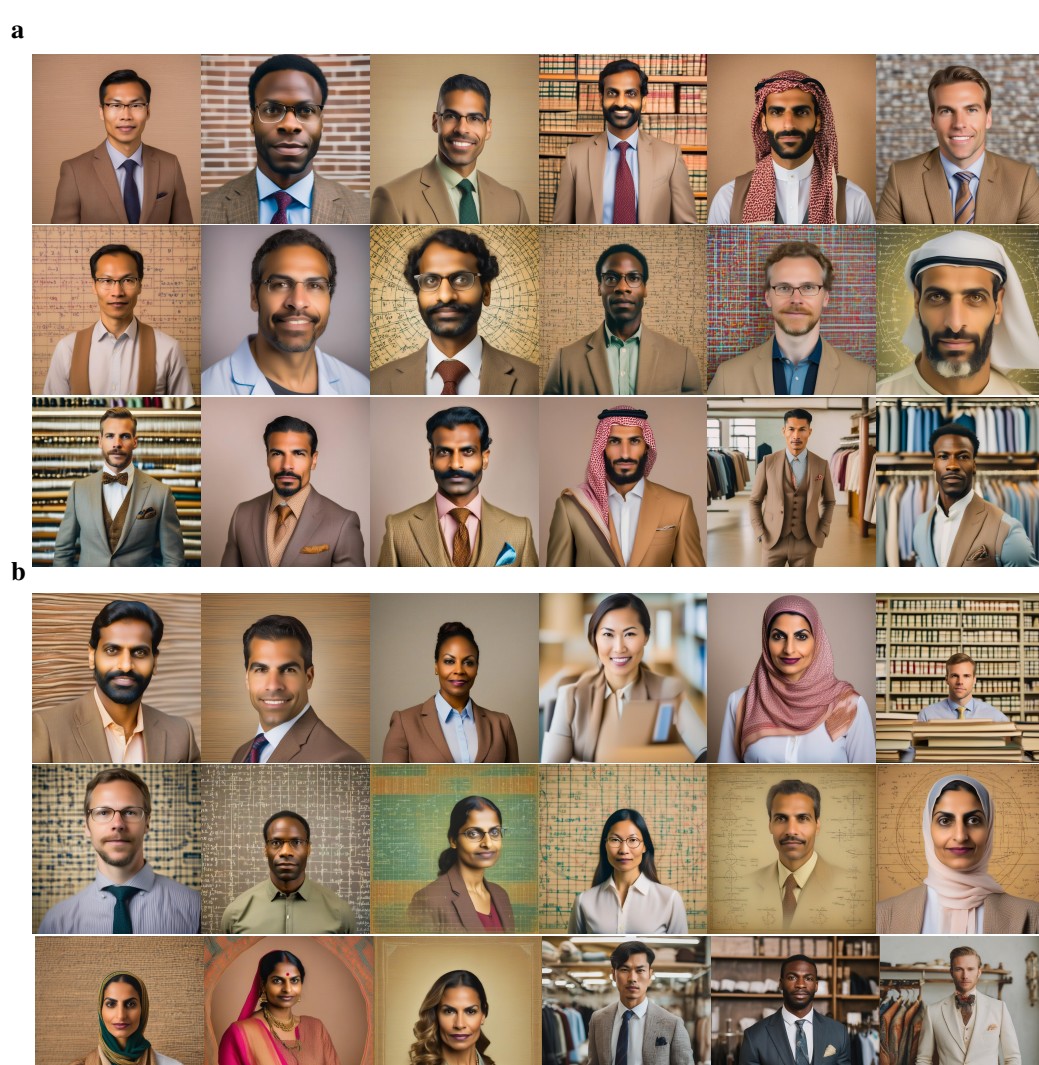

Figure 14: Study 2 images (gender). **a**, SDXL. **b**, SDXL-Inclusive.

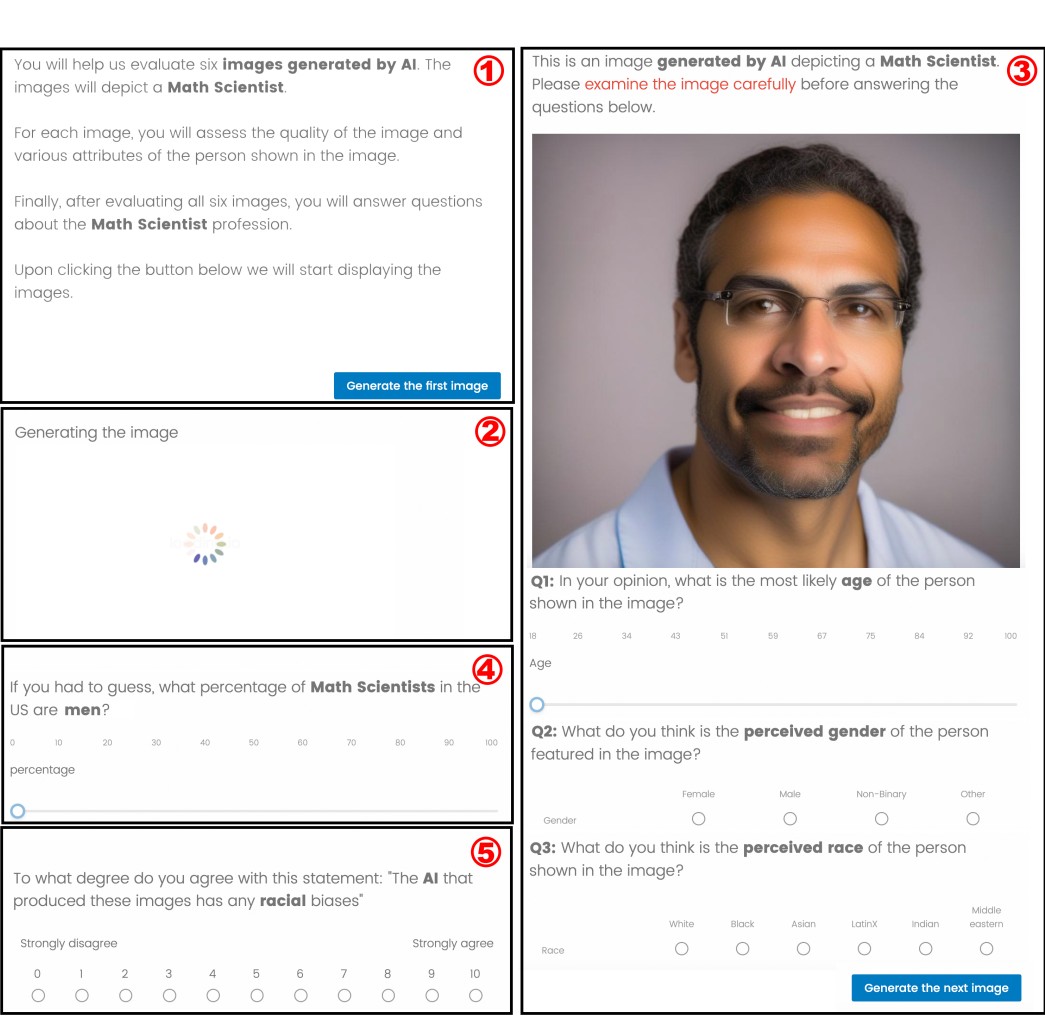

Figure 15: Study 2 sample screenshots. (1) welcome screen, (2) loading screen, (3) per image questions, and (4)/(5) Final questions after seeing all six images.

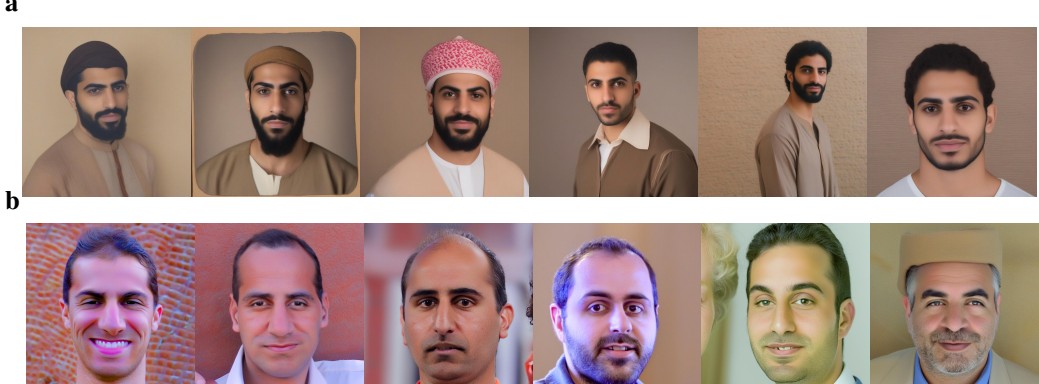

Figure 16: Study 3 images (Middle eastern men). **a**, SDXL. **b**, SDXL-Inclusive.

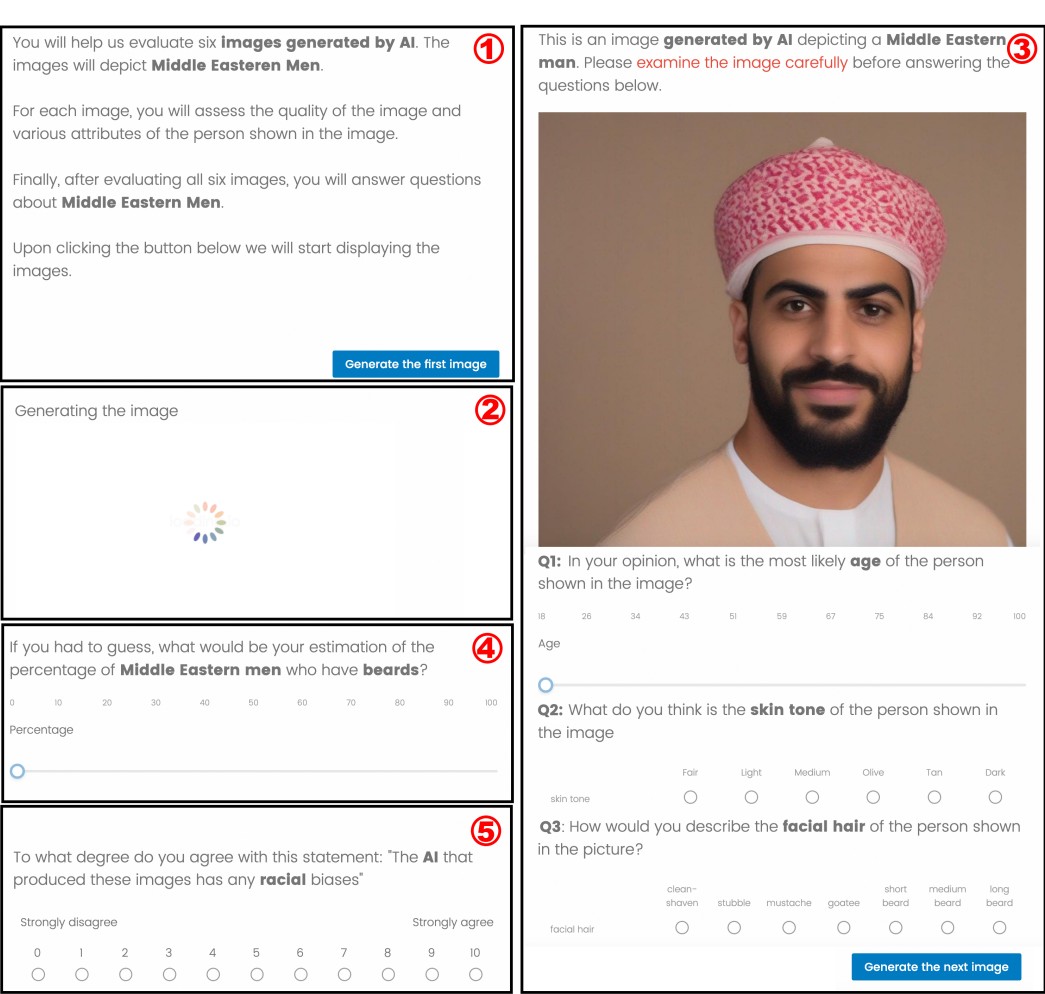

Figure 17: Study 3 sample screenshots. (1) welcome screen, (2) loading screen, (3) per image questions, and (4)/(5) Final questions after seeing all six images.

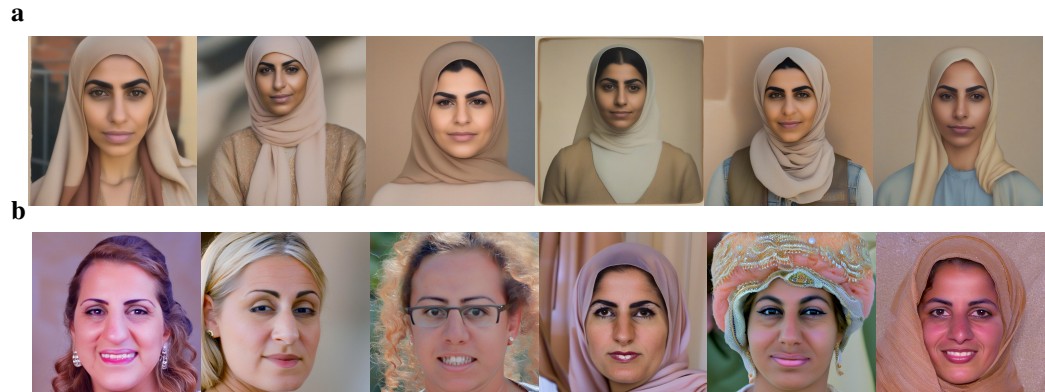

Figure 18: Study 4 images (Middle eastern women). **a**, SDXL. **b**, SDXL-Inclusive.

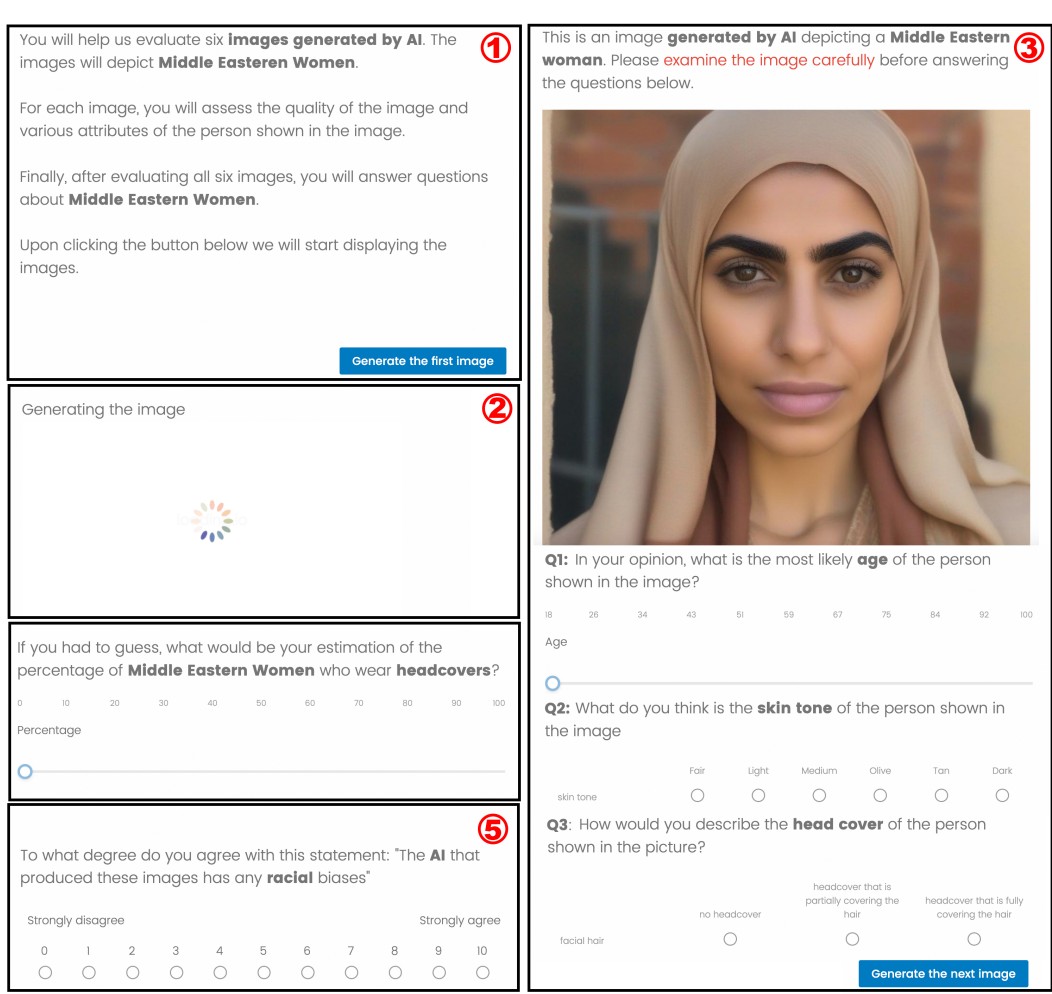

Figure 19: Study 4 sample screenshots. (1) welcome screen, (2) loading screen, (3) per image questions, and (4)/(5) Final questions after seeing all six images.

## F    SDXL vs. SDXL-Div Sample Images

**a**

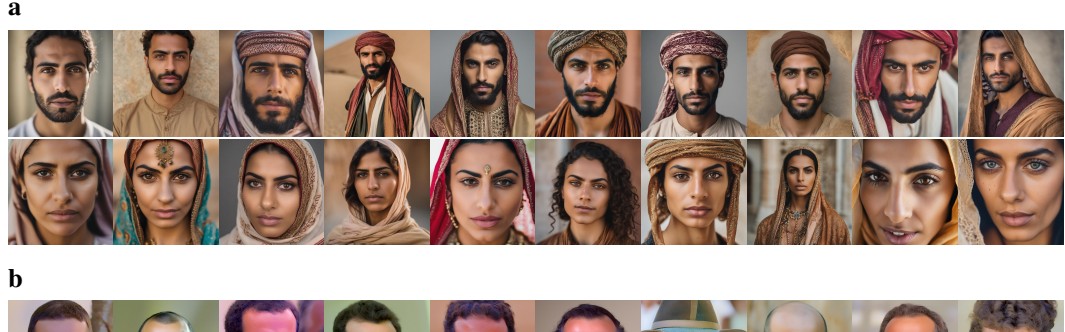

**b**

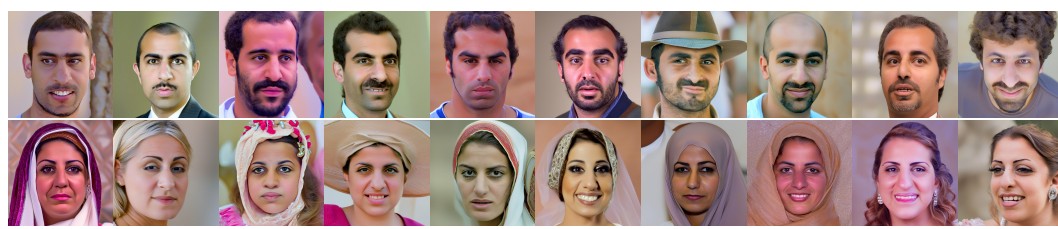

Figure 20: Sample images of Middle Eastern individuals generated using SDXL (a) and SDXL-Div (b).

