# OpenReview forum: "AI-generated faces influence gender stereotypes and racial homogenization"
_ICLR.cc/2025/Conference — Submitted to ICLR 2025_

### Official Review · Reviewer_TPme · 2024-10-31

**Soundness:** 3
**Presentation:** 1
**Contribution:** 2
**Rating:** 5
**Confidence:** 5

**Summary:**

The goal of this work is conducting an analysis of the stereotypes and biases present in Stable Diffusion XL (SDXL), a very well-known text-to-image generator used daily by a large amount of people around the world.
More specifically, the authors have, first of all, developed a state-of-the-art classifier of race, gender, professions from faces and have shown that the majority of faces generated by SDXL are white males as well as other biases (already found in the literature) such as the association between higher income and prestige jobs with white people. Another interesting finding is the racial homogenization present in the SDXL-generated images (for example, middle eastern men are depicted as dark-skinned and bearded while middle eastern women as wearing headscarf). Then, the authors have proposed a couple of debiasing solutions, SDXL-Inc to increase inclusivity and SXL-Div to decrease racial homogenization and increase diversity.
As last result, the authors conducted a series of user studies to investigate if the exposure to SDXL-Inc-generated faces can reduce racial and gender biases, while the exposure to SDXL-generated faces can increase these biases.

**Strengths:**

- interesting findings on the racial homogenization;
- extensive analyses of biases related to professions and attributes;
- interesting idea of user studies on the effect of exposure to inclusive vs. non-inclusive AI-generated faces.

**Weaknesses:**

The major issue with the paper is the organization. The paper targets a lot of different objectives and present a huge amount of findings but often presented and discussed in a quite superficial way. Overall, a lot of information is missing to better understand and evaluate the findings and also the methodologies adopted.
For example, what is done to create SDXL-Inc or SDXL-Div should be detailed more in order to understand the fine-tuning step (how is LORA used?).
Again, while only focusing on SDXL?
Moreover, the user studies are a very interesting part of the paper and I would give more focus to them. However, the number of study participants seems limited and it's not clarified at all this statement "we estimated that to obtain a power of 0.8 to detect a medium effect size (Cohen's d) of 0.5 in paired-sample comparison, a sample of 135 participants would be needed" ... could you better clarify this? More, how the study participants were selected on Prolific and details about them should be reported. Again, for sure it's interesting finding some short-term effect of exposure  to inclusive AI-generated faces but how long this effect lasts? and this effect should be better investigated for examples also with some follow-up interviews to study participants.
Again, more detail should be provided about ITI-GEN to better understand the comparison with SDXL-Inc. Overall, my suggestion is to focus the paper on less findings but going more in detail on presenting, validating and discussing them.

**Questions:**

The results obtained with SDXL-Inc have shown a significant increase of middle-eastern frequencies for all the profession while this is something not happening for Latinx. Do you have any explanation for this phenomenon? See for example Table 4 in the Appendix or Figure 7 always in the Appendix.

---

### Official Review · Reviewer_67Uj · 2024-11-01

**Soundness:** 2
**Presentation:** 1
**Contribution:** 1
**Rating:** 5
**Confidence:** 3

**Summary:**

The work reviews biases in Stable Diffusion XL (SDXL),  finding significant racial and gender stereotypes in generated images. The study finds that SDXL portrays some races with homogenized traits (e.g., Middle Eastern men as uniformly dark-skinned and bearded) and reinforces gender roles, depicting men in prestigious jobs and women in supportive roles. The authors propose SDXL-Inc for balanced racial and gender representation and SDXL-Div for enhancing diversity within racial groups. User studies show that exposure to inclusive AI-generated images can reduce biases, while non-inclusive content may increase them, stressing the importance of adopting inclusive AI practices.

**Strengths:**

The paper tackles an important and timely topic of fairness in AI-generated imagery, particularly addressing racial and gender stereotypes in a widely used model.

**Weaknesses:**

* Clarity: The drawbacks of previously proposed debiasing solutions remain unclear, even after reading the introduction and related work. The data overview feels like a list without clear guidance on how each dataset element should be interpreted. Additionally, SDXL-Inc is reintroduced redundantly in sections 3.2.1 and 4.2, which could be streamlined. The text is very dense. The results of the user studies are not explained. Why 32 professions?
* The paper would benefit from citing additional work on biases in text-to-image models, such as [1] and [2].
* More models should be evaluated as SDXL is an older model and only one among many current alternatives. Why did the authors focus on this one?
* Focusing on race can be problematic as it is a social construct and determining racial categories is often subjective, especially for AI-generated images. Assigning skin tone is typically more straightforward and less ambiguous.
* Novelty and significance: The paper does not compare the current solution with previously proposed debiasing methods. While characterizing biases in SD models has been done in past works, the paper needs to better emphasize what its unique contributions and novelties are.

**Questions:**

Questions are above.

---

### Official Review · Reviewer_gtJB · 2024-11-04

**Soundness:** 2
**Presentation:** 2
**Contribution:** 2
**Rating:** 3
**Confidence:** 4

**Summary:**

The authors investigate the biases of Stable Diffusion (SDXL) when generating faces with respect to 6 races, 2 genders, 32 professions, and 8 attributes. They further investigate how SDXL-generated images affect human biases about the representation of different races/genders in certain professions and racial homogeneity. They additionally study racial homogenization, i.e., the extent to which SDXL produces similar faces for individuals of the same race. The authors also propose debiasing methods called SDXL-Div and SDXL-Inc to combat homogenization and stereotyping by finetuning SDXL on diverse real and synthetic images of people of different races/genders.

**Strengths:**

- This work addresses two important issues that are underexplored in prior work: (1) measuring the homogeneity of SDXL depictions of racialized individuals, and (2) understanding how SDXL biases impact human biases. The authors’ finding that inclusive and diverse generations of different races/genders with respect to professions and physical appearance can reduce human biases is interesting. The authors could consider comparing the values reported in Figure 5 with, for example, values reported by the Bureau of Labor Statistics.

- The authors thoroughly discuss a subset of prior work and clearly highlight their research contributions in contrast to these papers. They reproduce the high-level findings of previous works, that Stable Diffusion generates disparate representations of individuals with different professions and attributes with respect to race and gender that reinforce hegemonic stereotypes and inequality.

- SDXL-Inc leads to a notable improvement in the measured representation of different races/genders. The authors’ finding that SDXL-Inc can improve racial/gender representation in generations for professions/attributes not seen during finetuning is interesting.

- The authors run their evaluations on 10,000 images per profession and per attribute.

**Weaknesses:**

- The authors’ theorization of race vs. skin color is unclear. For example, in lines 120-121, the authors state that by considering race, they “distinguish between, say, Asian and White individuals who happen to be equally light-skinned.” However, the racial categories considered are Western-centric and reductive. Furthermore, there is a considerable heterogeneity of skin tones among individuals in the same racial category. Additionally, the authors do not distinguish between racial identity vs. observed race vs. reflected race vs. racial roles [1].

- The authors’ discussion of related work, while deep, has limited breadth. For example, a primary claim is that few works propose automatic quantitative measures of bias in T2I systems and debiasing solutions. However, according to [2] (Appendices B and C), there appears to be a wealth of work that does so. It is further unclear how the authors’ proposed debiasing methods based on finetuning are novel and differ from existing approaches to debiasing based on finetuning (see Table 3 in [2]).

- In terms of presentation, the upfront description of all the datasets (especially datasets IV onward) yielded some confusion due to the reader not yet being familiar with the rest of the methodology. It is also difficult to refer back to the definitions of the different datasets as they are mentioned in the rest of the paper. Furthermore, many of the minute details (e.g., specific values of hyperparameters) can be introduced in the appendix for better readability.

- The authors’ proposed GPT-in-the-loop approach, wherein it is first automatically detected if an explicit race/gender is mentioned in a user prompt and if not, a random race/gender is injected into the prompt, is interesting, but may still not capture sufficient context to appropriately resolve biases (e.g., a random race/gender may not be desirable for historical image generations [3]). A future research direction could involve learning to take such context into account. The authors should also elaborate on Line 377; does the “in-the-loop” method do better than SDXL-Inc?

- [4] studies and explains the “bias amplification” phenomenon that the authors observe in Lines 260-267. Furthermore, the authors claim that finetuning SDXL on a dataset that is balanced with respect to gender and race will yield balanced gender and race representations (lines 280-283), but this appears to be in contradiction with the aforementioned “bias amplification” phenomenon.

- What are the demographics of the human participants? The race and gender of participants may affect their preconceptions of the representation of different races/genders in certain professions [5] and should be controlled for.

- Please use \citep instead of just \cite to ensure that inline citations are rendered correctly.

[1] https://aclanthology.org/2021.acl-long.149/

[2] https://arxiv.org/abs/2404.01030

[3] https://www.theverge.com/2024/2/21/24079371/google-ai-gemini-generative-inaccurate-historical

[4] https://aclanthology.org/2024.naacl-long.353/

[5] https://aclanthology.org/2023.acl-long.505/

**Questions:**

- Could the authors elaborate on the similarities and differences (if any) between “racial homogenization” and “stereotypes” as constructs?

- Why did the authors combine the East and Southeast Asian categories into a single category?

- What fairness issues do you envision arising from finetuning SDXL on the SDXL-Inc fine-tuning dataset, which entirely comprises images generated by SDXL? For example, do you foresee this amplifying existing biases [1]?

- What are the pros/cons of using automatic race/gender classification to measure biases in the depiction of individuals with certain professions, as opposed to more qualitative approaches (e.g., Average Face Comparison [2])?

- Lines 375-377: The authors note that the “in-the-loop” debiasing method also significantly reduces biases. What are the pros/cons of the “in-the-loop” method vs. finetuning 12 separate models?

- How do the authors validate that SDXL-Div produces high-fidelity generations? (Beyond just measuring if the embeddings of the generations have low cosine similarity.)

[1] https://dl.acm.org/doi/10.1145/3630106.3659029

[2] https://dl.acm.org/doi/10.5555/3666122.3668580

**Details Of Ethics Concerns:**

- Lines 133-139: The authors use a subset of LAION-5B, which was reported to contain CSAM [1].

- Lines 216-220: The authors use automatic classifiers for race and gender based on faces.

[1] https://cyber.fsi.stanford.edu/news/investigation-finds-ai-image-generation-models-trained-child-abuse

---

### Official Review · Reviewer_GLKg · 2024-11-04

**Soundness:** 2
**Presentation:** 2
**Contribution:** 2
**Rating:** 3
**Confidence:** 4

**Summary:**

The paper proposes debiasing AI models ( Stable Diffusion) by developing their classifiers with more inclusive datasets and then uses those inclusive images to assess whether people are biased or not, when shown AI generated images vs non AI generated images.

**Strengths:**

The authors have done an excellent job with literature review of prior research and similar research by their peers in similar areas of research.

**Weaknesses:**

Biases exist in humans whether one sees AI generated or non AI generated image, given that bias is a social and cognitive construct. I seriously missed the point of this research and what the authors wish to convey to the academic community with this. The datasets presented here such as FairFace etc have been shown by prior research to have been skewed towards certain races and genders and the authors' attempts to resolve them lack originality and poor applicability in real world applications.

**Questions:**

I would like the authors to throw more clarity on the underlying biases they wish to address both in people and the model. Especially here:  "Consequently,
it remains unclear whether such homogenization (if it exists) can be addressed by diversifying the
facial features of same-race individuals. Other open questions that have not been addressed to date
are whether being exposed to AI-generated faces can affect people’s racial and gender biases"

Also, I would like to see some literature review on how these particular attributes were selected in terms of social relevance when it comes to biases. The choice of two genders, 32 professions and 8 attributes - how is this representative of the population and why were they chosen?

---

### Meta-Review · Area_Chair_Qo58 · 2024-12-17

**Metareview:**

The paper proposes debiasing Stable Diffusion by developing classifiers with more inclusive datasets. To this end, the authors have curated a SoTA classifier for race, gender, profession from faces and show that the majority of faces generated by SDXL are white males and exhibit other biases including association between higher income and prestige jobs with white people. The authors propose SDXL-Inc for balanced racial and gender representation and SDXL-Div for enhancing diversity within racial groups. User studies show that exposure to inclusive AI-generated images can reduce biases, while non-inclusive content may increase them, stressing the importance of adopting inclusive AI practices.

The paper addresses important and timely, yet underexplored issues. The manuscript is well written, the breadth of literature review is impressive, and their contributions are clearly presented.

**Additional Comments On Reviewer Discussion:**

Representativeness. How two genders, 32 professions and 8 attributes may be representative of the population and why these were chosen is not clearly justified.

The authors’ theorization of race vs. skin color is unclear. Additionally, the authors do not distinguish between racial identity vs. observed race vs. reflected race vs. racial roles.

Clarity: The drawbacks of previously proposed debiasing solutions are unclear.

Paper organization: The paper targets a lot of different objectives and presents a large amount of findings but focusing on breath while lacking in depth.

The authors also did not address any of the questions raised by reviewers.

---

### Decision · Program_Chairs · 2025-01-22

Reject